# Isolated flat bands in 2D lattices based on a novel path-exchange symmetry

Jun-Hyung Bae[1], Tigran Sedrakyan[2] and Saurabh Maiti[1,3]

**1** Department of Physics, Concordia University, Montreal, QC H4B 1R6, Canada
**2** Department of Physics, University of Massachusetts, Amherst, MA 01003, USA
**3** Centre for Research in Molecular Modelling, Concordia University,
Montreal, QC H4B 1R6, Canada

## Abstract

The increased ability to engineer two-dimensional (2D) systems, either using materials, photonic lattices, or cold atoms, has led to the search for 2D structures with interesting properties. One such property is the presence of flat bands. Typically, the presence of these requires long-ranged hoppings, fine-tuning of nearest neighbor hoppings, or breaking time-reversal symmetry by using a staggered flux distribution in the unit cell. We provide a prescription based on carrying out projections from a parent system to generate different flat band systems. We identify the conditions for maintaining the flatness and identify a path-exchange symmetry in such systems that cause the flat band to be degenerate with the other dispersive ones. Breaking this symmetry leads to lifting the degeneracy while still preserving the flatness of the band. This technique does not require changing the topology nor breaking time-reversal symmetry as was suggested earlier in the literature. The prescription also eliminates the need for any fine-tuning. Moreover, it is shown that the subsequent projected systems inherit the precise fine-tuning conditions that were discussed in the literature for similar systems, in order to have and isolate a flat band. As examples, we demonstrate the use of our prescription to arrive at the flat band conditions for popular systems like the Kagomé, the Lieb, and the Dice lattices. Finally, we are also able to show that a flat band exists in a recently proposed chiral spin-liquid state of the Kagomé lattice only if it is associated with a gauge field that produces a flux modulation of the Chern-Simons type.



# 1   Introduction

The term 'flat band systems' has recently attracted a lot of attention. There are at least two contexts in which this term is used. The first, and probably the more popular, is in the context of Moiré bands [1,2] in twisted, layered Vanderwaals systems (epitomized by twisted bi-layer Graphene [3–9] and helical tri-layer Graphene [10]) where the multiple band-foldings due to enlarging of the unit-cell results in bands which can have significant regions in the Brillouin zone (BZ) where they disperse very weakly. This leads to an enhanced density of states, and if the chemical potential is around this region, many of the physics of itinerant electrons manifest themselves as a strongly correlated problem as the relevant dimensionless parameter $\nu_F U$ (where $\nu_F$ is the density of states at the Fermi surface and $U$ is some scale of interaction in the problem) could be made large even for small $U$. Another aspect driving the system towards strong correlation physics is the fact that the competition from the kinetic energy (which is characterized by the dispersiveness of a band) falls off due to the reduction of the bandwidth.

The second context in which the term 'flat band' is used is in the technically strict sense where systems have perfectly flat dispersionless bands. Some systems that are popularly discussed are the Kagomé lattice [11], the Lieb lattice [12], and the Dice lattice [13]. While there aren't any natural systems with these specific lattice structures, some of them can be realized in cross-sections of crystals [14], while some could be artificially engineered [15–19]. It should be noted that these systems have a perfectly flat band within the nearest neighbor (nn) approximation. Beyond the nn, the flatness is disturbed, of course, and the flat band acquires a bandwidth that is generally still much smaller than that of the 'flat bands' in the

Vanderwaals systems. In this work, we reserve the term 'flat' for this second context in the rest of this article.

Investigating flat bands is of fundamental interest [20] for a variety of reasons: it offers novel perspectives on topology [21, 22]; if they are topological, then it is expected that exotic physics of the fractional Quantum Hall effect could be observed in zero-field and at high temperatures [23]; a universal low-energy behavior that is different from the Fermi liquid can be expected as in the theory of the half-filled flat Landau level [24–26]; spin-liquid and chiral spin-liquid behaviors are associated with the presence of a flat energy manifold of excitations and serves as a platform to explore the role of Chern-Simons gauge field [27–32]; the presence of a flat band serves as a possible resolution to the fermion-doubling problem in lattice-based field theories [33]; to name a few. Perhaps the most intriguing yet achievable application of isolated flat band systems would be exploring the physics of the Sachdev-Ye-Kitaev (SYK) model [34–36]: e.g., introducing disorder leads to maximal chaos exhibiting black-hole like behavior (finite entropy at zero temperature) for which there are already various proposals for implementation [37–42]. However, many of these interesting effects only manifest themselves if the flat band is isolated (gapped) from the rest of the system. It is thus desirable to have a design prescription that achieves precisely this.

This desire has certainly been recognized by many. Investigation into the existence of the flat band itself revealed that the flat band would be degenerate with dispersive bands, with the degeneracy being protected by topology [43]. It was suggested in Ref. [44] that breaking Time Reversal Symmetry (TRS) was crucial to isolate the flat bands from the dispersive ones by considering staggered fluxes through the unit cell. In a sequence of works [45–47], it was demonstrated that breaking TRS was not necessary, but one would need to fine-tune the system using compact localized states for destructive interference of the electronic states, which would lead to a dispersionless band. There are general considerations from permutation symmetries in graph theory [48] and latent symmetries (associated with destructive interference across certain paths) [49] that can also explain the formation of flat bands on general grounds in general lattices. Certain special symmetry properties of the Hamiltonian (antiunitary-Parity-Time) can also lead to flat bands [50]. Reference [51] presented an interesting parameterization of the Kagomé lattice that also successfully isolated the flat band without the consideration of any special symmetries except what the authors identified as inversion [we shall show later (Sec. 5) that it does not have to do much with inversion]. Some works explore the possibility of having a flat band on general grounds, but they either require precise long-ranged hoppings [52] (which is not so desirable in material systems nor in photonic lattices nor ultra-cold atoms) or non-hermitian matrices [53]. Recently, flat bands have also been discussed in the context of hyperbolic lattices [54].

In this work, we add to the existing body of literature and show that it is possible to have isolated flat bands without breaking TRS, without using long-ranged hopping, without losing hermiticity, and without fine-tuning a system. At this point, we note that very recently in Refs [55,56], the authors presented a comprehensive analysis on the design of flat bands achieving the same goals of avoiding long-range hopping and fine-tuning. It provides an elegant and robust generalization of our results below, along with the topological classification of these bands. We encourage the reader to refer to this work for a more detailed analysis. While the fundamental requirements for both our works remain similar, we provide an alternative perspective through a focus on providing precise prescriptions to isolate a degenerate flat band and investigate the role of gauge fields in achieving the same. We start from a sufficient condition for the flatband to exist and arrive at a sufficient condition to isolate (gap out) the flatband. The distinguishing feature of our approach is that while the previous works focus on the properties of the Hamiltonian with a flat band, our method involves arriving at flat band systems by performing projections from a 'parent system.' In fact, we show that talking about

symmetries of the parent system allows for a simpler interpretation of the flat band in the projected systems. To emphasize this point, we first begin with the Kagomé lattice and show that naive attempts to isolate the flat band (via different onsite energies and applying strain) destroy the flat band. However, in addition to already existing prescriptions, we were able to identify another parameterization that preserves the flat band for all ranges of the parameter. But this technique (and the others discussed in the literature) often requires a very specific relationship between various hopping parameters. We then present our main result where we introduce a parent system with certain special properties that guarantees a flat band. And upon performing projections (to be detailed later in the text), we show that the projected systems automatically inherit the various conditions presented earlier for the existence of and isolation of the flat band.

We find this condition by first exploring bipartite systems with different system sizes and using the fact that such a system has the number of flat bands equal to the difference in the size of the subsystems [13], thus establishing a sufficient (but not necessary) condition to have flat bands. We then perform a Hilbert-space projection to project out the smaller subsystem, and we show that the larger subsystem will necessarily have flat bands. We explicitly demonstrate that our earlier parameterization of the Kagomé lattice and also some cases discussed earlier in the literature [51] are a special case of this projection prescription.

We also identify a symmetry associated with the bipartite system (and not the projected systems), which, when broken, isolates the flat band in the main system and its projected subsystems. We refer to this as a *path-exchange symmetry* with respect to a property which is the set of ratios of the paths (in this case hoppings) from different components of one subsystem to the other. If this set of ratios is the same for two atoms of the larger subsystem, then the whole system is said to have an exchangeable path between the subsystems. We show that the number of exchangeable paths directly controls the degree of degeneracy of the flat band with other dispersive bands. This more fundamental symmetry can map to spatial symmetries like mirror and inversion under special conditions and does not necessarily translate to something simple in the projected system and probably explains why there exist so many different attempts to understand the origin of the flat band in various systems. We then relax the bipartite condition in the subsystem that is being projected out and show that our main results still hold. We also show that the path-exchange symmetry argument holds beyond the nn approximation as long as the bipartite condition is maintained.

We also apply our prescription to the Lieb and Dice lattices and demonstrate the existence of other lattice structures where flat bands are present and can be isolated by breaking the path-exchange symmetry. In none of these cases where we isolate the flat band do we break time-reversal symmetry as was required in the staggered flux technique of Ref. [44]. Finally, as an application of our prescription, we are able to present scenarios where the hexagonal lattice (such as Graphene) or the chequerboard-like square lattice could also have a flat band. Since our prescription includes carrying out projections from a nn model, we believe that this prescription should be amenable to realization in photonic lattices and circuit QED systems [57–59].

As a closing example, we consider the case of a chiral spin-liquid state of fermionized bosons in Kagomé lattice subject to a Chern-Simon's (CS) field. It was proposed that this state has a flux distribution that would preserve the flat band as opposed to the situation with the regular (Maxwellian) flux which grows with the area [28]. We apply our results to the parent system with a regular Maxwellian flux and show that our projection procedure exactly reproduces the proposed CS flux distribution. The presence of the flat band in such a system was seen as a consequence of the characteristic flux distribution within the unit cell. From our work we now understand that the flat band is actually guaranteed from a more fundamental level.

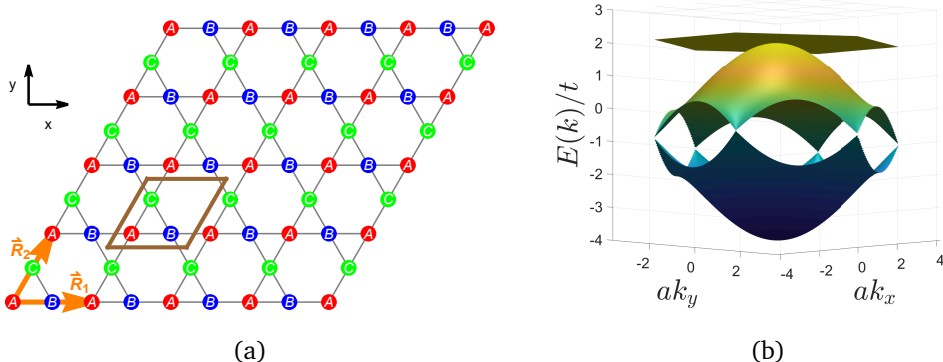

| (a) | (b) |

Figure 1: (a) Kagomé lattice with three atoms $A, B, C$ in the unit cell and the translation vectors $\vec{R}_1$ and $\vec{R}_2$. The brown parallelogram is a unit cell (b) The energy spectrum for the Kagomé lattice. Note the presence of the flat band that is degenerate with the dispersing middle band at the $\Gamma$-point (0,0).

The rest of the text is organized as follows. In Section 2 we introduce the Kagomé lattice and show that the common ideas to modify the lattice ends up disturbing the flat band. In Section 3, we introduce our parameterization that preserves the flat band and discuss the physical meaning of the parameterization. In Section 4 we introduce the projection prescription in terms of bipartite systems to generate flat bands. In Section 5 we identify the path-exchange symmetry in our bipartite systems, which upon being broken, isolates the flat band. We then show that the bipartite condition is not a strict requirement. In Section 6, we demonstrate all of the above ideas in the Lieb and Dice lattices. In Section 7 we discuss the validity of the proposed idea beyond the nn approximation. In Section 8, we demonstrate why the proposed Chern-Simons type flux distribution on a Kagomé lattice guarantees a flat band, whereas the usual Maxwellian flux distribution does not. Finally, we summarize our results in Section 9. The Appendix includes some details and proofs that did not find their place in the main text.

## 2 Kagomé: Naïve attempts to lift the flat band degeneracy

We start by considering the Hamiltonian for the Kagomé lattice within a tight-binding model with only nn hoppings:

$$H_{\text{Kg}} = -t \begin{pmatrix} 0 & 1+e^{-i\vec{k}\cdot\vec{R}_1} & 1+e^{-i\vec{k}\cdot\vec{R}_2} \\ 1+e^{i\vec{k}\cdot\vec{R}_1} & 0 & 1+e^{-i\vec{k}\cdot\vec{R}_3} \\ 1+e^{i\vec{k}\cdot\vec{R}_2} & 1+e^{i\vec{k}\cdot\vec{R}_3} & 0 \end{pmatrix}, \tag{1}$$

where $t$ is the nn hopping matrix element, $\vec{R}_1 = (1,0)$, $\vec{R}_2 = (\frac{1}{2}, \frac{\sqrt{3}}{2})$ are the translation vectors of the lattice, $\vec{R}_3 = \vec{R}_2 - \vec{R}_1$, and $\vec{k}$ is the Bloch momentum in the first Brillouin zone (fBZ) and is made dimensionless by absorbing the lattice constant $a$. This Hamiltonian is written in the basis $\hat{\Psi}_{\vec{k}} = (\hat{c}_{\vec{k},A}, \hat{c}_{\vec{k},B}, \hat{c}_{\vec{k},C})^T$, where $A, B, C$ are the three atoms within the unit cell [see Fig. 1(a)]. The spectrum contains a flat band as shown in Fig. 1(b).

In order to lift the degeneracy at the $\Gamma$-point, one could envision multiple ways to perturb the system: break the similarity of $A, B, C$ (sub-lattice symmetry), break or lower some translational or point-group symmetry, or break Time-reversal symmetry (TRS). We shall briefly demonstrate below that while the standard ways to apply these perturbations may lift the degeneracy at the $\Gamma$-point, they also destroy the flatness of the band. Further, the degeneracy point sometimes just gets moved to other points in the fBZ.

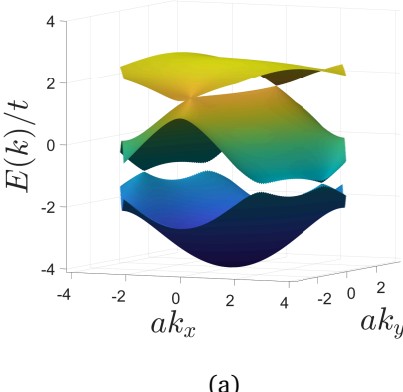
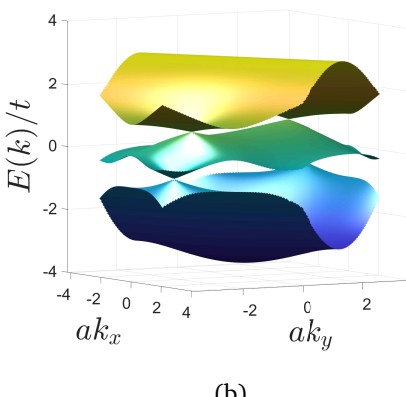

| (a) | (b) |

Figure 2: (a) Spectrum for the Kagomé system with different onsite energies ($\zeta = 0.5$). The $\Gamma$-point degeneracy is lifted, but it splits into two Dirac points at different $k$ values. The flatness of the flat band is also lost. (b) Spectrum for Kagomé lattice under a uniaxial strain along the diagonal that intersects the $B - C$ bond in Fig. 1a. Once again, the flat band is lost. Here $\delta t = 0.5$.

## 2.1 Onsite perturbations and strain

Consider first making the three atoms different by subjecting them to different onsite potentials. To model this, one could add the following term to the Hamiltonian: $\delta H_{\text{site}} = -t \, \text{Diag}(0, \zeta, -\zeta)$. For $\zeta \ll 1$, the $\Gamma$-point eigenvalues are $t\left(2 \pm \frac{\zeta}{\sqrt{3}}\right) + \mathcal{O}(\zeta^2)$ and $-4t + \mathcal{O}(\zeta^2)$. However, as seen in Fig. 2(a), numerical diagonalization shows that this condition just splits the quadratic $\Gamma$-point degeneracy into two Dirac points. So the degeneracy of the bands is not really lifted. This also destroys the flat band.

One can also consider modeling the effect of the strain. For simplicity, consider a uniaxial strain applied to the system. This would have two effects on the system: alter the translation vectors and alter the hoppings. The change in translation vectors will only act as a change in "gauge" in the $k$-space [60–63] and hence not really alter the qualitative aspects of the spectrum. The change in hoppings alters the symmetry properties of the $3 \times 3$ Bloch Hamiltonian. For the orientation shown in Fig. 1(a), applying a strain along the diagonal that intersects the $B - C$ bonds would result in the following Hamiltonian:

$$H_{\text{Kg,strain}} = -\begin{pmatrix} 0 & (t-\delta t)\left(1+e^{-i\vec{k}\cdot\vec{R}_1}\right) & (t-\delta t)\left(1+e^{-i\vec{k}\cdot\vec{R}_2}\right) \\ (t-\delta t)\left(1+e^{i\vec{k}\cdot\vec{R}_1}\right) & 0 & (t+\delta t)\left(1+e^{-i\vec{k}\cdot\vec{R}_3}\right) \\ (t-\delta t)\left(1+e^{i\vec{k}\cdot\vec{R}_2}\right) & (t+\delta t)\left(1+e^{i\vec{k}\cdot\vec{R}_3}\right) & 0 \end{pmatrix}. \quad (2)$$

The full spectrum plotted in Fig. 2(b) shows that the flat band is lost. Moreover, the quadratic touching point shifts away from the $\Gamma$-point as it splits into two Dirac points. This splitting of the $\Gamma$-point degeneracy into Dirac points is the same phenomenon that was discussed in Refs. [64, 65], although not in the context of strain.

## 2.2 Breaking TRS

In another attempt to lift the degeneracy of the flat band, we may also consider breaking TRS by applying an out-of-plane magnetic field to the system. Since we are working within a spinless model, the only effect of the magnetic field would be the orbital effect. We address this by constructing a Hofstadter model for the Kagomé lattice (see, e.g. [66]). In Fig. 3 we show, as

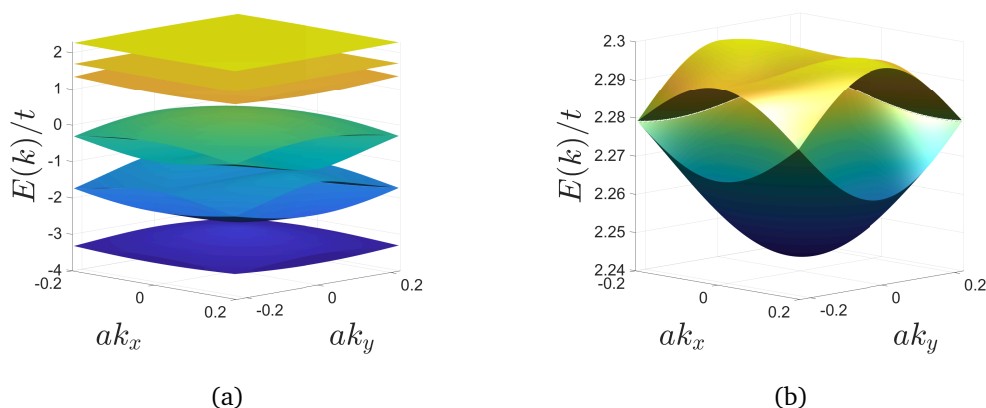

Figure 3: (a) Energy bands of the Kagomé lattice with $\pi\phi_0$ flux distributed uniformly in the unit cell. (b) A zoomed-in version of the top bands showing the dispersive nature. This remains true for any other uniform flux linked to the unit cell.

a representative case, the spectrum for a flux per unit cell of $\pi\phi_0$ (where $\phi_0 = \frac{h}{e}$ is the single electron flux quantum) and that the flat band becomes dispersive. There are, however, ways to break TRS and still preserve the flat band. This requires using staggered flux as introduced in Ref. [44] or a Chern-Simons flux (which is focused through one part of the unit cell) like in Ref. [28], and we shall return to this in Section 8.

## 3 Kagomé: Flat band preserving parameterization

While it seems that any perturbation we apply to the system lifts the flatness of the band, one may wonder if there could be a parameterization that preserves the flatness of the band. To tackle this question, let us consider the Kagomé Hamiltonian in the generic form

$$\mathcal{H} = -t \begin{pmatrix} 0 & \alpha_1 & \alpha_2 \\ \alpha_1^* & 0 & \alpha_3 \\ \alpha_2^* & \alpha_3^* & 0 \end{pmatrix}. \tag{3}$$

The eigenvalues are the roots of the equation:

$$-(E/t)^3 + (E/t)\left(|\alpha_1|^2 + |\alpha_2|^2 + |\alpha_3|^2\right) - 2\text{Re}\left[\alpha_1^* \alpha_2 \alpha_3^*\right] = 0. \tag{4}$$

To have a flat band at $E/t = f$ (independent of $\vec{k}$), we necessarily need the $\vec{k}$-dependence of $\alpha_i$ to be such that for all $\vec{k} \in$ fBZ

$$|\alpha_1|^2 + |\alpha_2|^2 + |\alpha_3|^2 = \frac{2\text{Re}\left[\alpha_1^* \alpha_2 \alpha_3^*\right]}{f} + f^2. \tag{5}$$

In fact, if such a flat band were to exist, then Eq. (5) would have to be satisfied for some real value of $f$. Plugging this into the characteristic equation leads us to

$$-(E/t)^3 + (E/t)\left(\frac{A}{f} + f^2\right) - A = 0, \tag{6}$$

where $A \equiv 2\mathrm{Re}\left[\alpha_1^* \alpha_2 \alpha_3^*\right]$. The eigenvalues would then be

$$E_0 = tf \,,$$
$$E_+ = \frac{tf}{2}\left(-1 + \sqrt{1 + 4A/f^3}\right),$$
$$E_- = \frac{tf}{2}\left(-1 - \sqrt{1 + 4A/f^3}\right). \tag{7}$$

If there exists no such $f$, then the above expressions are no longer the solution. One could then ask what are the conditions for which $f$ could exist or equivalently, under what conditions would Eq. (5) be satisfied. Note that this equation presents one constraint on the parameters of this equation which are $\alpha_i$ ($i \in \{1, 2, 3\}$). At this stage, it might seem like one should always be able to satisfy this. However, note that the $\alpha_i$'s are, in turn, functions of $k_x, k_y$ and $f$. The $k_i$'s are subject to the condition that they must be bound to the fBZ. Given that the $k_i$'s appear as arguments of periodic functions, this bound is not really an additional constraint. But, the $\alpha_i$'s themselves are not independent. In fact, the Kagomé Hamiltonian's structure ensures that $(\alpha_1 - 1)^*(\alpha_2 - 1) = \alpha_3 - 1$, which are two more constraints on the parameters. Thus, we have a situation with 3 constraints and 3 parameters. Since they are not linear, they may or may not be satisfied in general.

For completeness, we could ask if any of the dispersive bands ($E_\pm$) could intersect the flat band $E_0$. Setting them equal to each other immediately establishes that for real values of $A$ and $f$, only $E_+$ could intersect with $E_0$ and this would happen at those $\vec{k}$-points where

$$2\mathrm{Re}\left[\alpha_1^* \alpha_2 \alpha_3^*\right] \equiv A = 2f^3 \,. \tag{8}$$

In fact, for the case of the Kagomé lattice, we note that $\alpha_i = 1 + e^{-i\vec{k}\cdot\vec{R}_i}$ and Eq. (5) is satisfied for $f = 2$ and for all $\vec{k}$, establishing the condition for the flat band. Further, Eq. (8) is also satisfied only at the $\Gamma$-point indicating that the flat band would be degenerate with the dispersive band at the $\Gamma$-point.

One may now wonder if there are other scenarios, other than the Kagomé lattice, where Eq. (5) could be satisfied. The answer is affirmative and a family of scenarios is presented below. Consider the modification where $\alpha_i$ is changed from $1 + e^{-i\vec{k}\cdot\vec{R}_i}$ to $(1 + r) + (1 - r)e^{-i\vec{k}\cdot\vec{R}_i}$, with $|r| < 1$. The physical meaning of the $r$-parameter will be discussed in a subsequent section. In fact, one could factor out $1 + r$ and absorb it into the hopping element $t$ yielding the following transformation $t \to \tilde{t} = t(1 + r)$, and

$$\alpha_i \to \tilde{\alpha}_i = 1 + \frac{1 - r}{1 + r}e^{-i\vec{k}\cdot\vec{R}_i} = 1 + e^{-i\vec{k}\cdot\vec{R}_i - h} \,,$$

where $h$ is defined through the equation $r = \tanh\frac{h}{2}$. This allows us to express

$$\tilde{\alpha}_i = 2\cos\left(\frac{\vec{k}\cdot\vec{R}_i}{2} - \frac{ih}{2}\right)e^{-i\frac{\vec{k}\cdot\vec{R}_i}{2} - \frac{h}{2}} \,.$$

From these definitions, we observe that the flat band condition of Eq. (5)

$$\sum_{i=3}|\tilde{\alpha}_i|^2 = \frac{2\mathrm{Re}[\tilde{\alpha}_1^* \tilde{\alpha}_2 \tilde{\alpha}_3^*]}{f} + f^2 \,, \tag{9}$$

can now be satisfied with $f = 2e^{-\frac{h}{2}}\cosh\frac{h}{2}$ (see Appendix A for details). The spectrum is given by Eq. (7) but with $\alpha_i \to \tilde{\alpha}_i$ and $t \to \tilde{t}$. The condition for band-touchings also remains the same as Eq. (8) but with $\alpha_i \to \tilde{\alpha}_i$. A direct evaluation shows that if the condition is satisfied

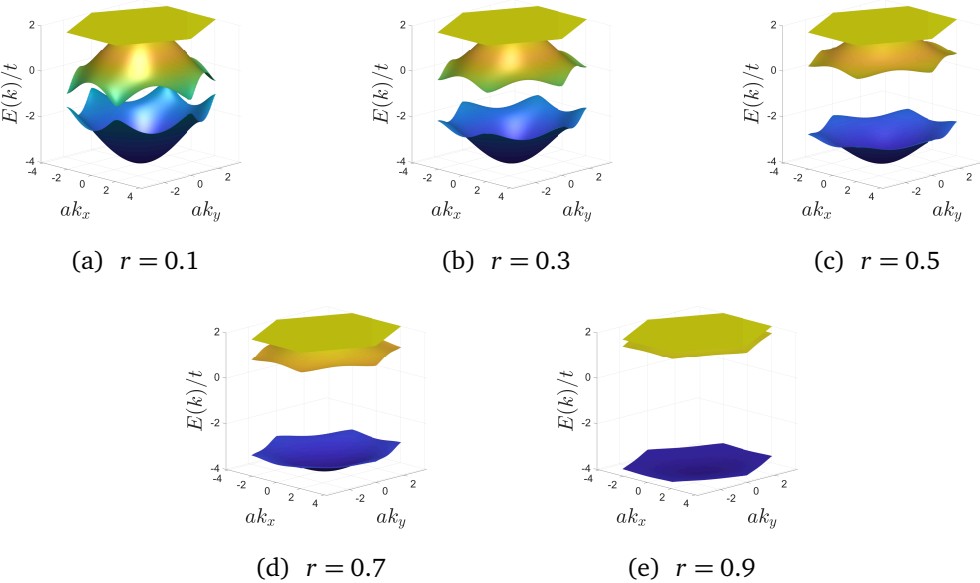

Figure 4: Evolution of the spectrum for the Kagomé lattice with the parameter $r$. The flat band is located at $E = \tilde{t}f = 2t$. As $r$ increases, the Dirac points gap out, but the flat band maintains a quadratic band touching with the dispersive band. As $r \to 1$, the dispersive band merges with the flat band. The spectrum remains the same under $r \to -r$.

for $\alpha_i$, then it is automatically satisfied for $\tilde{\alpha}_i$. This means that the flat band remains flat at $E = \tilde{t}f = 2\tilde{t}e^{-\frac{h}{2}}\cosh\frac{h}{2} = 2t$ . The introduction of the parameter $r$ neither lifts nor moves the degeneracy at the $\Gamma$-point. In fact, observe that when $r \to 0$, $\tilde{\alpha}_i \to \alpha_i$, and the formulae smoothly connect to the original Kagomé lattice. Thus, we have a family of flat band systems characterized by the parameter $r$.

The spectra for the modified lattices are shown in Fig. 4 for various values of $r$. As observed above, for any value of $r$, the flat band is preserved. Although the Dirac point at the $K$ and $K'$ points of the fBZ are gapped out, the $r$ parameter preserves the degeneracy at the $\Gamma$-point consistent with the analysis above.

## 3.1 Physical meaning behind the $r$-parameter

The introduction of the $r$ parameter allowed for the same parameterization of the flat band condition as the original Kagomé lattice but with a modified (complex) phase. Further, this parameterization also allows us to interpret $t(1+r)$ as hopping within the unit cell and $t(1-r)$ as hopping outside the unit cell (because this is the term in the Bloch Hamiltonian associated with the translation phase factor). For $r > 0$, it would correspond to bringing the 3 atoms closer together (without altering the lattice constant) towards one corner of the unit cell, as shown in Fig. 5. For $r < 0$ the deformation takes the atoms toward the other corner of the unit cell. This deformation is nothing but the 'breathing' anisotropy considered in frustrated Kagomé lattices [67]. Further, the case with $r > 1$ corresponds to negative $t_{inter}$, which is equivalent to having a $\pi$ phase attached to the hopping element. This case will be discussed in more detail in the next subsection.

Observe from Fig. 5 that when $r = 1$, the intercell hopping $t(1-r) = 0$ (molecular limit) and we do not hop to the neighboring unit cells and thus we get a non-dispersive 3-level system. Because of the non-dispersive nature, the bands are trivially flat, and two of them are

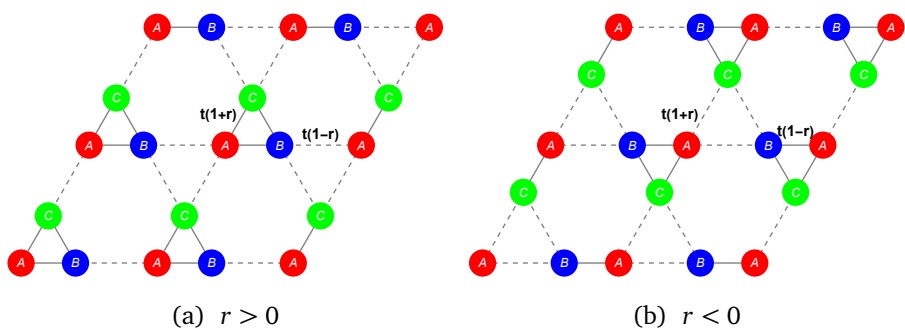

$$(a) \ r > 0 \qquad\qquad (b) \ r < 0$$

Figure 5: Modified Kagomé lattice. For $r > 0$, the basic modification is such that it brings the atoms together towards one corner of the unit cell. For $r < 0$ the atoms are moved closer toward the other corner of the unit cell. The original Kagomé lattice is restored at $r = 0$.

degenerate because we are still satisfying Eq. (8) for touching of the flat band with one other band. For $r < 0$, we effectively swap intercell and intracell hoppings. This just amounts to mirroring the unit cell about a diagonal and thus does not change anything in the spectrum.

## 3.2 Model with $|r| > 1$

When $|r| > 1$ one of either the inter-cell hopping or the intra-cell hopping parameters goes negative. From a solid-state point of view, this is clearly unphysical, however, we still have a well-defined spectrum and eigenfunctions. One could imagine the bonds with negative hoping to be associated with a phase of $\pi$. In Fig. 6 (a) we show the resulting flux distribution. This is clearly a $2\pi$-flux per unit cell, but the flux is modulated within the unit cell such that the flux is concentrated in the hexagonal region and one of the two triangular regions in the unit cell. This could be viewed as the flux through the closed structure in the unit cell (the triangle ABC) having zero flux, and the flux is only "in between" these closed structures. In the limit $r \to \infty$ (recall that the energy of the flat band is independent of $r$) the lattice returns to the Kagomé form factor, albeit with the modified flux. Although there is flux distribution within the unit cell, TRS is still preserved because the phase is $\pi$ which is the same as $-\pi$. In fact, this is exactly one of the cases arrived at in Ref. [44] but is naturally included in our parameterization.

The spectrum for $|r| > 1$ is shown in Fig. 6 (b). As discussed in an earlier subsection, at $r = 1$, the dispersing band merges with the original flat band. However, as $r$ becomes greater than 1, the dispersing band moves above the flat band. This is an interesting example where one band completely passes through another one. Note also that the $\Gamma$-point remains degenerate.

Another application of models with $|r| > 1$ would be in scenarios where a coupled system of oscillators is mapped to the tight-binding model. This is realizable in photonic-crystal systems and even in cold atom systems. Thus, we have demonstrated that there exists a parameterization in terms of the change of the basis of the unit-cell, mathematically realized by introducing the parameter $r$, that preserves the flat band. At this stage, this is just one additional means to discuss the condition for flat bands. However, we shall now describe a prescription to generate flat bands on general grounds, which includes all the cases discussed above (and in the literature).

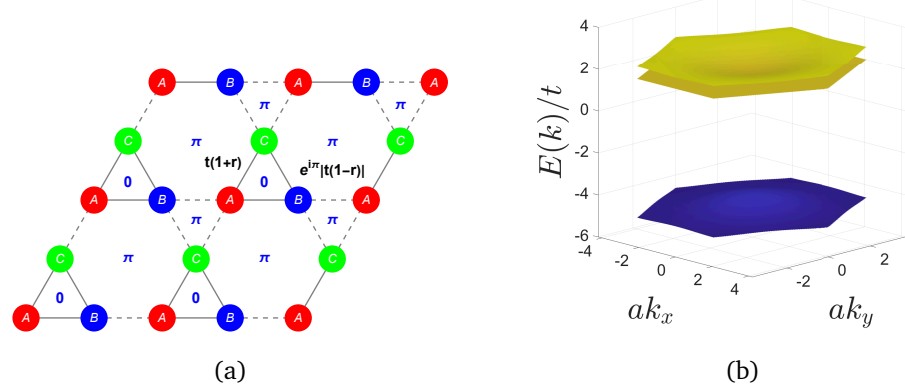

(a)           (b)

Figure 6: (a) Modified Kagomé lattice for $r > 1$. The negative hopping could be seen as the bond having a $\pi$-phase that results in characteristic $2\pi$-flux distribution per unit cell as shown. (b) The energy spectrum for $r = 1.2$. The flat band is preserved and so is the degeneracy at the $\Gamma$-point. However, note that the dispersive band is now above the flat band.

# 4 A prescription to generate flat band systems

In the previous sections, we presented a detailed analysis of the Kagomé system and its modifications that would preserve the flat band. The approach was rather direct where we searched for the parameters that would keep a band dispersionless ($\vec{k}$-independent). This approach, however, is not generalizable to investigate other systems as they would have to be dealt with on a case-by-case basis. However, the presence of the flat band in the Kagomé lattice could be deduced in a rather interesting manner. Consider a bipartite system as shown in Fig. 7 with 2 and 3 atoms per unit cell and hoppings only between the respective subsystems: consisting of X, Y atoms and A, B, and C atoms. The Hamiltonian is given by

$$
H_5 = -t \begin{pmatrix}
0 & 0 & 1 & 1 & 1 \\
0 & 0 & 1 & e^{-i\vec{k}\cdot\vec{R}_1} & e^{-i\vec{k}\cdot\vec{R}_2} \\
1 & 1 & 0 & 0 & 0 \\
1 & e^{i\vec{k}\cdot\vec{R}_1} & 0 & 0 & 0 \\
1 & e^{i\vec{k}\cdot\vec{R}_2} & 0 & 0 & 0
\end{pmatrix},
\tag{10}
$$

where the basis is $\hat{\Psi}_{\vec{k}} = (\hat{c}_{\vec{k},X}, \hat{c}_{\vec{k},Y}, \hat{c}_{\vec{k},A}, \hat{c}_{\vec{k},B}, \hat{c}_{\vec{k},C})^T$. There is a well known property of a bipartite Hamiltonian, $H_{(n+m)\times(n+m)}$ (of subsystem sizes $n$ and $m$), one can always construct the matrix $\mathcal{C} \equiv \text{Diag}(1_{n\times n}, -1_{m\times m})$ such that $\{\mathcal{C}, H\} = 0$ (i.e. it anti-commutes with the Hamiltonian). This implies that for every state with energy $E$, there must exist another orthogonal state at energy $-E$. This is commonly known as a particle-hole symmetric spectrum. Fig. 7(b) shows the numerically evaluated spectrum for $H_5$. It is easy to argue from this property that if $m + n$ is odd, we need to have a state at $E = 0$ and thereby guaranteeing a flat band [13]. However, we wish to show that there is a more fundamental reason for having flat bands which goes beyond this standard argument.

Starting from a bipartite system, consider projecting out one subsystem, by using the Löwdin's method [68], at some energy scale of interest $E_0$. This is a method that projects out one subspace from the system and the scale $E_0$ plays the role of chemical potential in most cases. This projection for the bipartite system is a special case of a more general mathematical construction involving $2^n$-root topology [72]. Let us suppose that we would like to project out

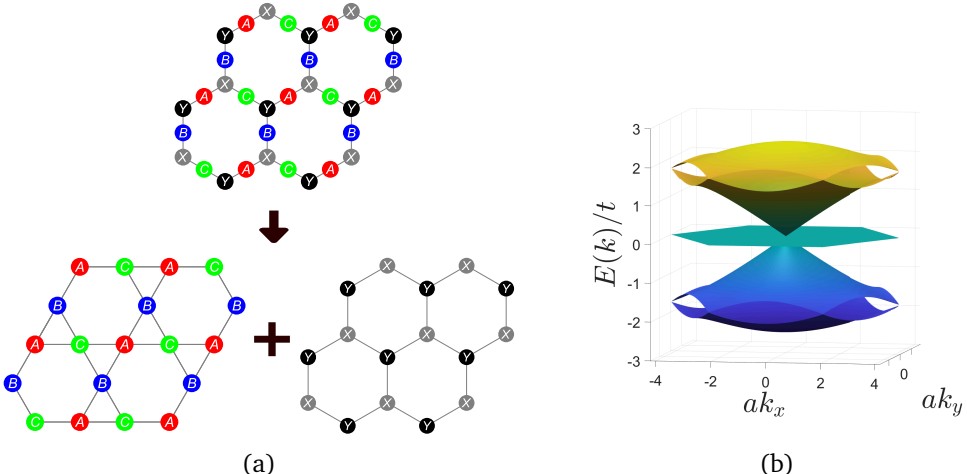

Figure 7: (a) Bipartite system with subsystems 2 and 3 atoms per unit cell. (b) The energy spectrum for the 5 atoms per unit cell structure is particle-hole symmetric. This is guaranteed by the bipartite nature of the system.

the subsystems A, B, and C and express the Hamiltonian purely in terms of the states of the other subsystems X, Y. To apply the Löwdin's method, first view the Hamiltonian $H_5$ as blocks

$$\begin{pmatrix} [H_{GG}]_{2\times2} & [H_{GK}]_{2\times3} \\ [H_{KG}]_{3\times2} & [H_{KK}]_{3\times3} \end{pmatrix}.$$

Then, the effective Hamiltonian projected onto the $G$ space (consisting of X,Y) would be

$$H_{\text{eff,G}}(E_0) = H_{GG} + H_{GK}[E_0 - H_{KK}]^{-1}H_{KG}. \tag{11}$$

Because of the bipartite nature with $K \to$ Kagomé and $G \to$ Graphene, $H_{GG} = 0$ and $H_{KK} = 0$. This yields

$$H_{\text{eff,G}}(E_0) = \frac{H_{GK}H_{KG}}{E_0}. \tag{12}$$

Similarly, the effective Hamiltonian projected onto the $K$ space is

$$H_{\text{eff,K}}(E_0) = \frac{H_{KG}H_{GK}}{E_0}. \tag{13}$$

Usually, Löwdin's method is used in the perturbative sense at some energy scale that separates out the states far away from that energy scale. However, due to the bipartite nature, we can perform an exact projection, as outlined above. There are two observations of interest for the Hamiltonians $H_{\text{eff,G}}$ and $H_{\text{eff,K}}$:

1. We can recognize

$$H_{\text{eff,G}} = \frac{t^2}{E_0}\begin{pmatrix} 3 & 1 + e^{i\vec{k}\cdot\vec{R}_1} + e^{i\vec{k}\cdot\vec{R}_2} \\ 1 + e^{-i\vec{k}\cdot\vec{R}_1} + e^{-i\vec{k}\cdot\vec{R}_2} & 3 \end{pmatrix}, \tag{14}$$

as the Hamiltonian for Graphene and

$$H_{\text{eff,K}} = \frac{t^2}{E_0}\begin{pmatrix} 2 & 1 + e^{-i\vec{k}\cdot\vec{R}_1} & 1 + e^{-i\vec{k}\cdot\vec{R}_2} \\ 1 + e^{iR_1} & 2 & 1 + e^{-i\vec{k}\cdot\vec{R}_3} \\ 1 + e^{i\vec{k}\cdot\vec{R}_2} & 1 + e^{i\vec{k}\cdot\vec{R}_3} & 2 \end{pmatrix}, \tag{15}$$

as the Hamiltonian for the Kagomé lattice. This could have been expected since the second order hops connects each subsystem to itself.

2. Note that $H_{KG} = H_{GK}^\dagger$. A non-square matrix $M$ has the property that $M^\dagger M$ and $MM^\dagger$, which are of different ranks, share the same eigenvalues, with additional zeroes making up for the difference in ranks (see Appendix B). Thus, our two subsystems will be such that $H_{\text{eff,K}} \sim H_{KG}H_{GK}$ will have the *same eigenvalues* as $H_{\text{eff,G}} \sim H_{GK}H_{KG}$ but an additional 0 due to the rank mismatch. This zero is $\vec{k}$ independent and hence results in a flatband. This is the more fundamental reason behind the formation of flat bands and most other conditions, if not all, are derivable from this construction. We demonstrate a few other cases later in this article.

This explains why the Kagomé lattice spectrum has a flat band and also why the rest of the spectrum is exactly the same as Graphene.

From point (2) above, we can conclude a general rule that if one constructs a bipartite system, projecting out the smaller subsystem will result in a flat band in the new effective system. The generality of the prescription outlined above implies that the detailed structure (the dimensions or even the matrix elements) of $H_{GK}$ does not matter. In fact, our $r$ parameterization is a special case of this general rule. To demonstrate this point, consider first, the $H_5$ Hamiltonian where

$$H_{GK} = -t \begin{pmatrix} 1 & 1 & 1 \\ 1 & e^{-i\vec{k}\cdot\vec{R}_1} & e^{-i\vec{k}\cdot\vec{R}_2} \end{pmatrix}.$$

This can be generalized to

$$H_{GK} = -\begin{pmatrix} t_{G_1 K_1} & t_{G_1 K_2} & t_{G_1 K_3} \\ t_{G_2 K_1} & t_{G_2 K_2} e^{-i\vec{k}\cdot\vec{R}_1} & t_{G_2 K_3} e^{-i\vec{k}\cdot\vec{R}_2} \end{pmatrix},$$

and the flat band would still persist owing to the size mismatch of the two subsystems. The $r$ parameterization (and all of the ensuing discussion) corresponds to the choice of $t_{G_1 K_i} = \sqrt{t(1+r)}$ and $t_{G_2 K_i} = \sqrt{t(1-r)}$, for $i \in \{1, 2, 3\}$, and hence preserves the flat band. We emphasize that the prescription we provide (based on bi-partiteness) is a *sufficient* condition to get a flat band but not a necessary one.

Observe that in the projected subsystems the flat band remains at zero energy. In a bipartite system with a flat band, the chiral symmetry guarantees particle-hole symmetric spectrum. The eigenvalues of the projected system are simply square of the parent system. This means that the two subsystems have the same eigenvalues besides the flat band [55], which is inherited by one of these subsystems. Since the square is non-negative definite and the flat band is at zero energy, it follows that the projection procedure ensures that the flat band is always at the extremities. This will be explicitly evident in several examples that we discuss below.

## 4.1 Flat bands beyond the bipartite condition

A bipartite system with different system sizes having a flat band is a rather straightforward result. However, we now wish to show that this condition is not necessary. We can start from the bipartite system and then allow for hoppings or energies in the subsystem to be projected out. This would still guarantee the presence of the flat band (the particle-hole symmetry will no longer persist, of course) in the parent system and also in the projected subsystem (with the larger size). The existence of the flat band is controlled solely by the existence of the non-square coupling matrix between a subsystem that does not talk to itself and another subsystem with a smaller size.

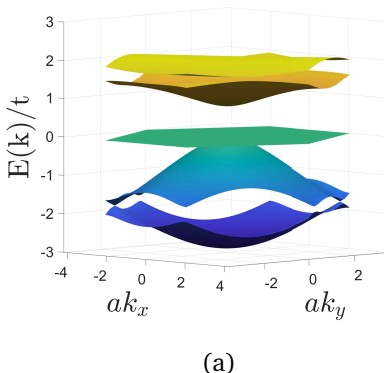

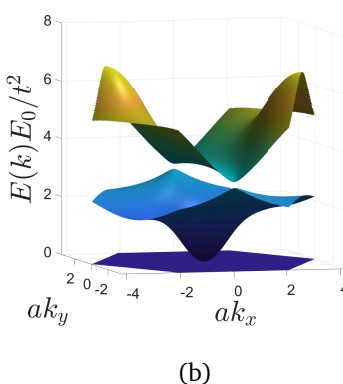

(a)                                   (b)

Figure 8: (a) The spectrum of the $H_5$ system still has a flat band with $H_{GG} \neq 0$ (but $H_{KK} = 0$), despite the violation of the bipartite condition. (b) The flat band in the projected subsystem.

To see this, let us first note that the effective Hamiltonian now goes from

$$H_{KG} H_{GK} \rightarrow H_{KG} [E_0 - H_{GG}]^{-1} H_{GK} \,,$$

where $H_{GG}$ is an arbitrary Hermitian matrix. The matrix $E_0 - H_{GG}$ can be diagonalized as $M \Lambda M^\dagger$, where $\Lambda$ is the diagonal matrix of the eigenvalues of $H_{GG}$ and $M$ is the matrix of eigenvectors of $H_{GG}$. Thus, $[E_0 - H_{GG}]^{-1} = M[E_0 - \Lambda]^{-1} M^\dagger$. This allows us to write

$$H_{KG} [E_0 - H_{GG}]^{-1} H_{GK} = \underbrace{H_{KG} M}_{\tilde{H}_{KG}} [E_0 - \Lambda]^{-1} \underbrace{M^\dagger H_{GK}}_{\tilde{H}_{GK}} \,.$$

Due to Hermiticity we have $H_{GK} = H_{KG}^\dagger$ and this implies $\tilde{H}_{GK} = \tilde{H}_{KG}^\dagger$. Thus, we arrive at the form $[\tilde{H}_{KG}]_{ac} D_{cd} [\tilde{H}_{GK}]_{db}$, where $D_{ab} = d_a \delta_{ab}$ is a diagonal matrix. Since, $\tilde{H}_{KG} = \tilde{H}_{GK}^\dagger$, the above product can be written as $[\bar{H}_{KG}]_{ac} [\bar{H}_{GK}]_{cb}$, where $[\bar{H}_{KG}]_{ab} = [\tilde{H}_{KG}]_{ab} \sqrt{d_a}$. Since this maintains $\bar{H}_{KG} = \bar{H}_{GK}^\dagger$, we can map this non-bipartite system to a strict bipartite system and apply all the same arguments: the size mismatch of the two systems would result in a zero eigenvalue and hence a flat band. In this sense, we only need the bipartiteness to choose the subsystems and then allow the smaller subsystem to have any hoppings and the resulting system would still have a flat band. We demonstrate this in Fig. 8 where we include the off-diagonal elements in $H_{GG}$ as $t'(1 + 0.1 e^{i \vec{k} \cdot \vec{R}_1} + 0.5 e^{i \vec{k} \cdot \vec{R}_2})$ and its c.c with $t' = 0.5 E_0$ (this simply accounts some hoppings in the GG subspace). The particle-hole symmetry is lost, but the flat band still exists. Note that this is also highlighted in Ref. [55].

Lastly, we show that deviation from the stated condition above destroys the flat band. Consider the simple case of adding different onsite energies to the sites of the subsystem of interest. Consider

$$H_{KK} = \begin{pmatrix} E_a & 0 & 0 \\ 0 & 0 & 0 \\ 0 & 0 & 0 \end{pmatrix} \,.$$

The spectrum for such a case is shown in Fig. 9(a) and the ensuing projected system also shown in (b) also loses the flatness. As another example for demonstrating that violating the stated condition destroys the flatness, consider the strained Hamiltonian in Eq. (2) which does

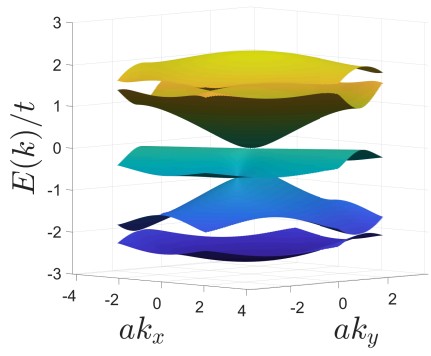

(a) $H_5$ with onsite energy

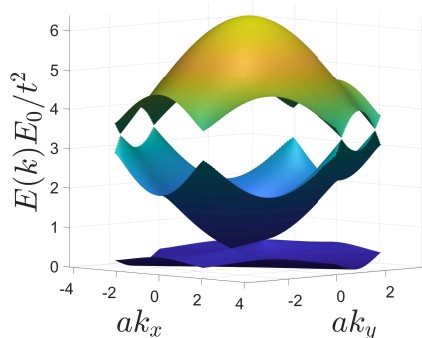

(b) Projected Kagomé with on-site energy

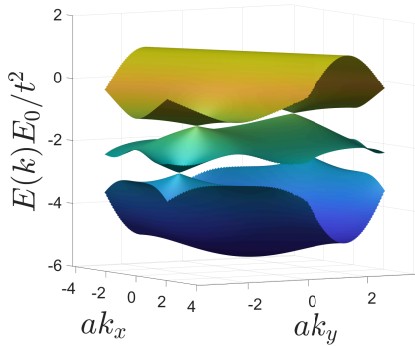

(c) Projected Kagomé with non-bipartite terms

Figure 9: (a) The energy spectrum for the $H_5$ system with an on-site energy $E_a = t$. (b) The spectrum after projection from the $H_5$ system onto the ABC subsystem. (c) The spectrum after projection from the $H_5$ system with the non-bipartite terms added to the ABC subsystem. Here, $\delta t = 0.5t$. In both cases, ($b$) and ($c$), the loss of our stated flat band condition indeed results in the loss of the flat band.

not have a flat band. This is so because arriving at this form from the $H_5$ formulation requires choosing

$$H_{KK} = \begin{pmatrix} 0 & c_1 & c_2 \\ c_1^* & 0 & c_3 \\ c_2^* & c_3^* & 0 \end{pmatrix}, \tag{16}$$

with $c_1 = -\frac{t\delta t}{E_0}(1+e^{-i\vec{k}\cdot\vec{R}_1})$, $c_2 = -\frac{t\delta t}{E_0}(1+e^{-i\vec{k}\cdot\vec{R}_2})$, $c_3 = \frac{t\delta t}{E_0}(1+e^{-i\vec{k}\cdot\vec{R}_3})$. This violates our stated condition that $H_{KK}$ still needs to be of the 'bipartite' nature. Indeed after Löwdin's projection we get

$$H_{\text{eff}} = H_{\text{eff,K}} + H_{KK}, \tag{17}$$

where $H_{\text{eff,K}}$ is the same as in Eq. (15). This is nothing but the strained Hamiltonian of Eq (2), with an overall scale factor of $\frac{t}{E_0}$. The spectrum is shown in Fig. 9(c). Physically including $c_i$ amounts to including nnn terms in the $H_5$ system.

# 5 Isolating the flat band

Having formulated a technique to generate a family of flat band systems, we note that in the cases we looked at, the flat band always appeared degenerate with a dispersive band. However,

if we let $t_{G_i K_j}$ to be different from each other, we preserve the flat band (since any change within the matrix $H_{GK}$ is allowed) as well as isolate it from the dispersive band. As an example, consider the case where we displace one of the $a, b,$ or $c$ atoms such that it is closer to $x$ and further from $y$ in Fig. 10(b) such that with $t_{G_1 K_1} = t_{G_1 K_2} = t_{G_2 K_1} = t_{G_2 K_2} = t, t_{G_2 K_3} = t + \delta t$ and $t_{G_1 K_3} = t - \delta t$. After projecting the system onto the Kagomé form, an explicit calculation shows that the gap at the $\Gamma$-point is $\frac{\delta t^2}{3t}$ (for $\delta t \ll t$).

**The path-exchange symmetry:**  The above change falls under the case shown in Fig. 10(d). This differs from the cases in Fig. 10(a), (b), and (c) in the following way. Consider the sets of paths taking us from the smaller subsystem ($XY$) to a given element in the larger one ($ABC$): $\{XA, YA\}, \{XB, YB\}, \{XC, YC\}$. Consider then, the set $\{r_X^{AB}, r_Y^{AB}\}$ created from the ratios $r_X^{AB} \equiv \frac{XA}{XB}$ and $r_Y^{AB} \equiv \frac{YA}{YB}$. The elements of this set correspond to the ratio of paths-to-larger-subsystem from the atoms of the smaller subsystem. Special situations arise when all the entries of the set are identical (i.e., can be reduced to a unit set). When we find a pair AB where the set of ratios could be reduced to a unit set, then we say we have a path-exchange in the system (physically, the presence of a unit set between pairs of atoms means that the paths to hop across subsystems are exchangeable between these two atoms). We show in Appendix C that for a bipartite system (of size $m, n$ with $m > n$), the presence of $z$ such path-exchanges leads to a $2(z - [m-n]) + 1$-fold degeneracy of the flat band with dispersive bands. The parameter $z$ is simply the number of reducible rows of the Hamiltonian. Sometimes, at certain points in the Brillouin zone, the Hamiltonian may simply have null rows. In such a case we cannot construct the path-exchange interpretation, but the identification with the number of reducible rows still holds.

Returning to Fig. 10, note that in (a) our sets are $\{\frac{XA}{XB}, \frac{YA}{YB}\} \rightarrow \{1, 1\}$ for AB; $\{\frac{XA}{XC}, \frac{YA}{YC}\} \rightarrow \{1, 1\}$ for AC; and $\{\frac{XB}{XC}, \frac{YB}{YC}\} \rightarrow \{1, 1\}$ for BC. Since both B and C can be exchanged with A, we say that there are *two* unique path-exchanges and hence $z = 2$. This leads to a triple degeneracy $[2(z - [m-n]) + 1 = 3]$ in the $H_5$ system. In (b) the ratios for the same pairs are also $\{1, 1\}, \{1, 1\},$ and $\{1, 1\}$, despite the hoppings being different. In (c), the ratios are $\{1, 1\}, \{\alpha, \alpha\},$ and $\{\alpha, \alpha\}$, where $\alpha \neq 1$. Since all the sets could be reduced to unit sets, we still have three identical path and hence two path-exchanges and the degeneracy remains three-fold. And finally in (d) the ratios are $\{1, 1\}, \{\beta, 1\},$ and $\{\beta, 1\}$. Since there is only one unit set possible this time, there is only one exchangeable path and $z = 1$. This lowers the degree of degeneracy to 1 (basically lifting the degeneracy) as shown in Fig. 10(e) and (f).

**Note:**  In general, with hopping parameters $t_{G_i K_j}$, the projected Hamiltonian in the Kagomé subspace is

$$H_{\text{eff,K}} = \frac{1}{E_0} \begin{pmatrix} t_{G_1 K_1}^2 + t_{G_2 K_1}^2 & t_{G_1 K_1} t_{G_1 K_2} + t_{G_2 K_1} t_{G_2 K_2} e^{-i\vec{k}\cdot\vec{R}_1} & t_{G_1 K_1} t_{G_1 K_3} + t_{G_2 K_1} t_{G_2 K_3} e^{-i\vec{k}\cdot\vec{R}_2} \\ c.c. & t_{G_1 K_2}^2 + t_{G_2 K_2}^2 & t_{G_1 K_2} t_{G_1 K_3} + t_{G_2 K_2} t_{G_2 K_3} e^{-i\vec{k}\cdot(\vec{R}_2 - \vec{R}_1)} \\ c.c. & c.c. & t_{G_1 K_3}^2 + t_{G_2 K_3}^2 \end{pmatrix}. \tag{18}$$

This form is the same as in Ref. [51] where the authors decomposed their Kagomé Hamiltonian into contributions from 'upper' triangles and 'lower triangles'. In their interpretation, in order to isolate the flat band they hypothesized breaking inversion (in the bonds) in the unit cell. This is not technically correct. It is not sufficient to break inversion at the level of the hoppings between the nearest neighbor atoms: the changes in hoppings need to be correlated in a definite manner which was only evident because of the author's chosen parameterization. One example of inversion broken deformation of the Kagomé unit cell is the $r$-parameterization discussed in Sec. 2. This parameterization breaks inversion but does not lift the flat band degeneracy. To understand why, consider the $H_5$ Hamiltonian with the parameterization above

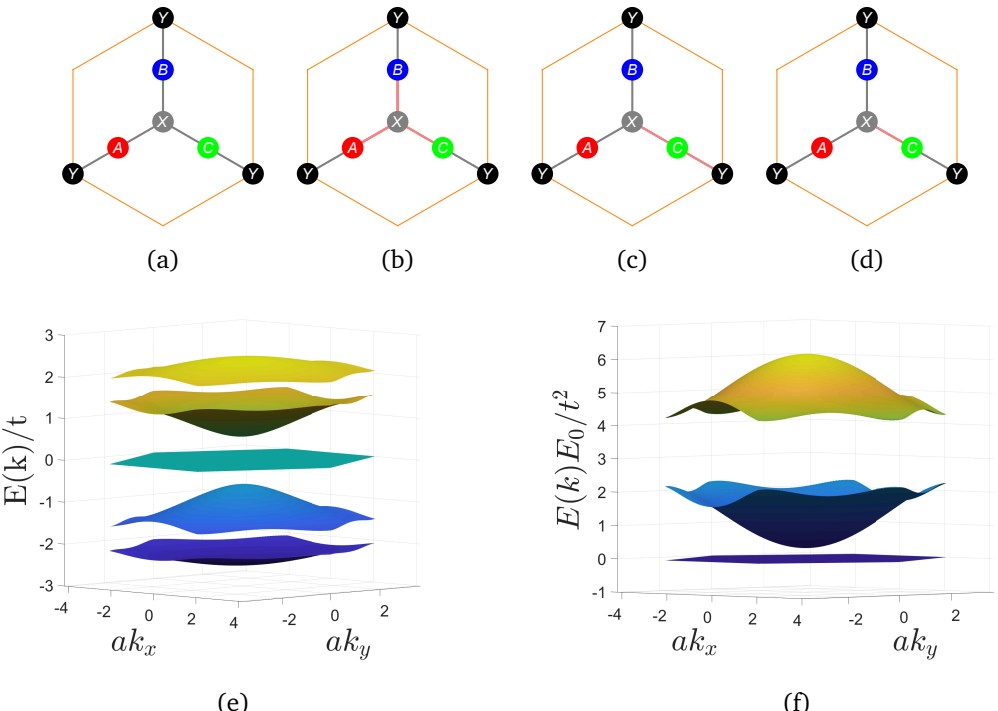

Figure 10: Hoppings in the unit cell of the $H_5$ system. The pink bonds are different from the grey bonds. The situations in (a), (b), and (c) do not lift the flat band degeneracy but that in (d) does. This is because, in the former three, the paths for individual atoms for going from the ABC subsystem to XY subsystem are exchangeable. Whereas in (d) that condition is broken. (e) Isolated flat band in the spectrum of the exchange-broken $H_5$ system and (f) the same for the projected Kagomé subsystem with the exchange-breaking perturbation being $\delta t = 0.5t$. The flat band is preserved and isolated from the dispersing bands.

where $t_{G_i K_j} = t$ except $t_{G_2 K_3} = t + \delta t$ and $t_{G_1 K_3} = t - \delta t$. This results in the following projected Hamiltonian in the Kagomé subspace:

$$
H_{\text{eff,K}}(\Delta) = \frac{t^2}{E_0} \begin{pmatrix} 2 & 1 + e^{-i\vec{k}\cdot\vec{R}_1} & (1-\Delta) + (1+\Delta)e^{-i\vec{k}\cdot\vec{R}_2} \\ 1 + e^{i\vec{k}\cdot\vec{R}_1} & 2 & (1-\Delta) + (1+\Delta)e^{-i\vec{k}\cdot(\vec{R}_2 - \vec{R}_1)} \\ (1+\Delta)e^{i\vec{k}\cdot\vec{R}_2} & (1+\Delta)e^{i\vec{k}\cdot(\vec{R}_2 - \vec{R}_1)} & 2 + 2\Delta^2 \end{pmatrix}, \quad (19)
$$

where $\Delta = \frac{\delta t}{t}$. Notice that in the Kagomé subspace, this isn't just an arbitrary breaking of the mirror as the $\Delta$ has to enter the onsite energy in precisely the stated manner to preserve the flatness of the band. However, in the $H_5$ system, it is sufficient to break the path-exchange symmetry in the bonds, in any manner possible with no other conditions, and we arrive at the appropriate Hamiltonian in the projected basis with all the necessary conditions built-in.

In this regard, we emphasize that while the statement in Ref. [51] that their stated perturbation that isolates the flat band also breaks inversion is correct, we wish to state that breaking inversion does not necessarily gap out the flat band and that inversion may not have anything to do with the problem at hand. This is evident from the $r$-parameterization presented earlier which also breaks inversion but preserves the degeneracy of the flat band.

We note in passing that this path-exchange symmetry is a fundamental symmetry of bipartite graph. Certain spatial symmetries such as mirror, rotations, inversions can be a special case of this symmetry. It is thus easy to associate the spatial symmetries to the cause of degeneracies. This is not incorrect, but they are not fundamental. In the examples we present in this

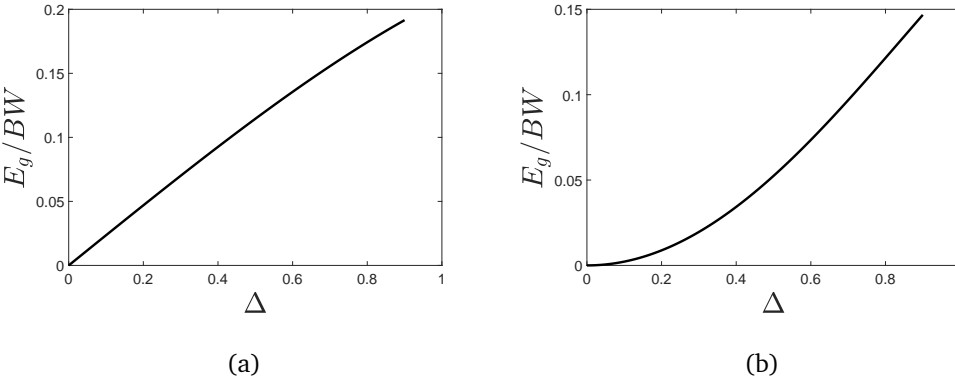

Figure 11: The band gap between the flat band and the nearest dispersing band relative to the full band width for the parent $H_5$ system (a) and for the projected system (b), with respect to the path-exchange breaking parameter $\Delta$.

article, most of the path-exchange symmetries can be broken by breaking a mirror symmetry or a $C_n$ symmetry in the physical lattice.

## 5.1 Evolution of the flat band gap

It is informative to quantify the gap that opens up due to the exchange-symmetry breaking. In the $H_5$ system the gap opens up at the $\Gamma$ point, which can be evaluated by solving for the eigenvalues at $\vec{k} = (0,0)$. Consider the following exchange-broken Hamiltonian $H_5$ at the $\Gamma$-point:

$$H_5^\Delta(\Gamma) = t \begin{pmatrix} 0 & 0 & 1 & 1 & 1+\Delta \\ 0 & 0 & 1 & 1 & 1-\Delta \\ 1 & 1 & 0 & 0 & 0 \\ 1 & 1 & 0 & 0 & 0 \\ 1+\Delta & 1-\Delta & 0 & 0 & 0 \end{pmatrix}. \tag{20}$$

The flat band remains at 0 and the other four eigenvalues are calculated as

$$E = \pm t \sqrt{3 + \Delta^2 \pm \sqrt{9 - 2\Delta^2 + \Delta^4}}, \tag{21}$$

which gives the relative gap size ($E_g$) with respect to the full bandwidth ($BW$) as

$$\frac{E_g}{BW} = \frac{\sqrt{3 + \Delta^2 - \sqrt{9 - 2\Delta^2 + \Delta^4}}}{2\sqrt{3 + \Delta^2 + \sqrt{9 - 2\Delta^2 + \Delta^4}}}. \tag{22}$$

For $\Delta \ll 1$, $\frac{E_g}{BW} \approx \frac{\Delta}{3\sqrt{2}}$. This means that the gap scales linearly with the symmetry-breaking perturbation. In the projected system, however, since the eigenvalues are square of that in the parent system, we will have $E_g^{\text{Proj}} \approx \frac{4\Delta^2 t^2}{9} E_0$, where $E_0$ is the energy scale of projection and $\frac{E_g^{\text{Proj}}}{BW} \approx \frac{2\Delta^2}{9}$. Due to the quadratic dependence, the gap only manifests far larger values of the perturbation. The evolution of the gaps in the parent and projected systems is shown in Fig. 11.

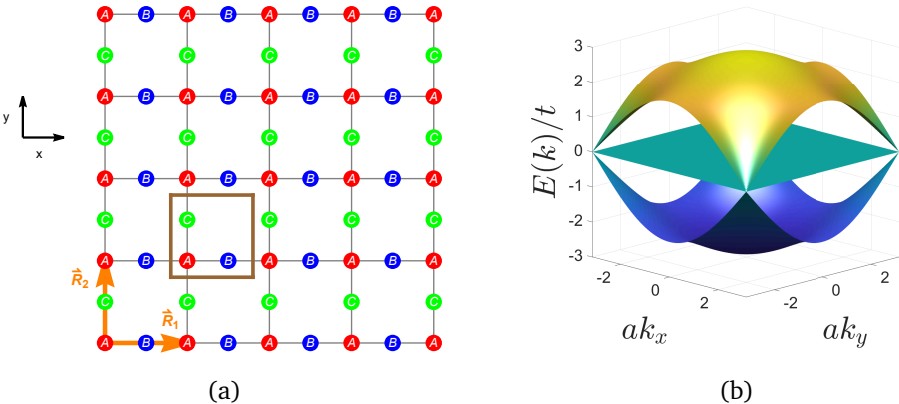

Figure 12: (a) Lieb lattice with three atoms $A, B, C$ in the unit cell and the translation vectors $\vec{R}_1$ and $\vec{R}_2$. (b) The energy spectrum for Lieb lattice. Note the presence of the flat band at the center of the particle-hole symmetric spectrum.

## 6 Application of the prescription to other lattices

Thus far we derived the well-known lattices like the Kagomé and Graphene systems as projections from a parent $H_5$ system and indicated that isolating the flat band requires breaking the path-exchange symmetry in their parent system. We now demonstrate that the other systems that are discussed in the literature (like Lieb and Dice) are actually the parent systems to other flat-band lattices and that the flat band could be isolated by breaking the path-exchange symmetry of the parent system. We shall only show the results for the strict bipartite case to keep the discussion simple, but as we have shown, this is not needed.

### 6.1 Lieb lattice and its projections

The Lieb lattice shown in Fig. 12(a) is another realizable example of the prescription. The system is bipartite within the nn approximation and the Hamiltonian is given by

$$
H_{\text{Lb}} = -t \begin{pmatrix} 0 & 1+e^{-i\vec{k}\cdot\vec{R}_1} & 1+e^{-i\vec{k}\cdot\vec{R}_2} \\ 1+e^{i\vec{k}\cdot\vec{R}_1} & 0 & 0 \\ 1+e^{i\vec{k}\cdot\vec{R}_2} & 0 & 0 \end{pmatrix},
\tag{23}
$$

where all the terms have an analogous meaning as in Eq. (1), but in this case $\vec{R}_2 = (0, 1)$. Here, however, the Lieb lattice is the analog of the $H_5$ Hamiltonian, prior to projection. Hence, the spectrum is particle-hole symmetric as shown in Fig. 12(b). Carrying out the projections using the Löwdin's method, the two subsystems are

$$
H_{\text{eff,Sq}}(E_0) = \frac{2t^2}{E_0} \left[ 2 + \cos\left(\vec{k}\cdot\vec{R}_1\right) + \cos\left(\vec{k}\cdot\vec{R}_2\right) \right],
\tag{24}
$$

which is the regular square lattice; and the other Hamiltonian is

$$
H_{\text{eff,xSq}}(E_0) = \frac{t^2}{E_0} \begin{pmatrix} 2+2\cos(\vec{k}\cdot\vec{R}_1) & 1+e^{i\vec{k}\cdot\vec{R}_1}+e^{-i\vec{k}\cdot\vec{R}_2}+e^{i\vec{k}\cdot(\vec{R}_1-\vec{R}_2)} \\ 1+e^{-i\vec{k}\cdot\vec{R}_1}+e^{i\vec{k}\cdot\vec{R}_2}+e^{-i\vec{k}\cdot(\vec{R}_1-\vec{R}_2)} & 2+2\cos(\vec{k}\cdot\vec{R}_2) \end{pmatrix},
\tag{25}
$$

which is also a square lattice with 2-atoms per unit cell, which we may refer to as the extended square lattice. The lattices and spectrum of these two effective systems are shown in Fig. 13. It

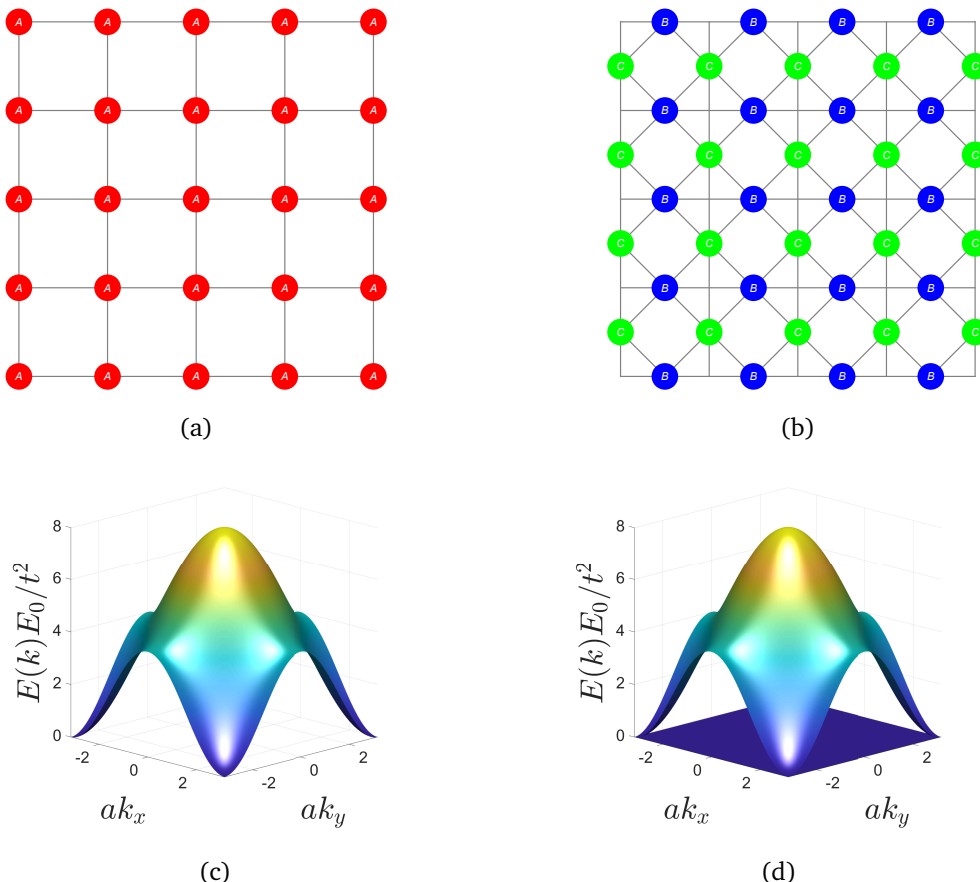

Figure 13: (a) and (b) The two subsystems (square and extended square lattices) from projecting out the Lieb lattice. (c) and (d) The energy spectrum for the respective subsystems. Note the presence of the flat band in the extended square lattice.

is worth noting that without the projection technique, one would need to know the exact ratios of the nn and nnn hoppings to ensure the presence of the flat band in the two band system. However, this technique ensures that the projected systems already have the appropriate ratios to have the flat band.

Further, like we identified $H_{GK}$ in $H_5$, one can identify the matrix $H_{SX}$ with

$$H_{SX} = \left(1 + e^{-i\vec{k}\cdot\vec{R}_1} \quad 1 + e^{-i\vec{k}\cdot\vec{R}_2}\right), \tag{26}$$

which can be generalized to

$$H_{SX} = \left(t_{AB} + \tilde{t}_{AB}e^{-i\vec{k}\cdot\vec{R}_1} \quad t_{AC} + \tilde{t}_{AC}e^{-i\vec{k}\cdot\vec{R}_2}\right). \tag{27}$$

We can now break the path-exchange symmetry in the Lieb lattice [see Fig. 14(b)] by selecting the parameterization $t_{AC} = (t - \delta t)$ and $\tilde{t}_{AC} = (t + \delta t)$. We show in Fig. 14(d) the spectrum of the exchange-broken Lieb lattice projected onto the extended square lattice system which shows the isolated flat band.

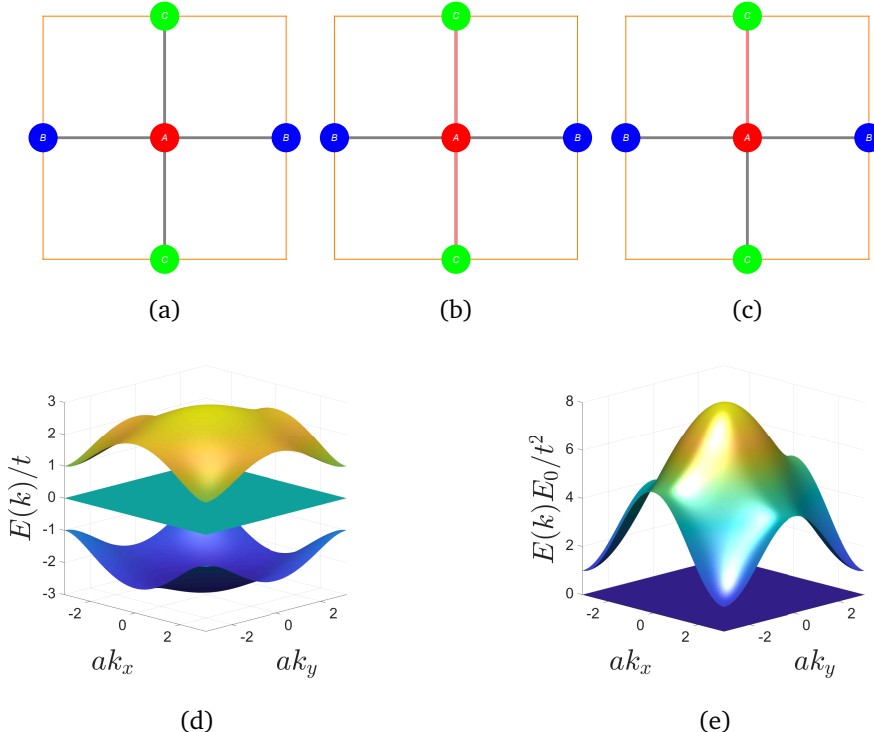

Figure 14: Hoppings in the unit cell of the Lieb lattice with A being one system and BC being the other. In situations (a) and (b), where the path-exchange between the subsystems is preserved, the flat band degeneracy is not lifted degeneracy, but it is in (c) where the condition is broken. (d) Isolated flat band in the spectrum of the exchange-broken Lieb lattice and (e) the same but projected onto the extended square lattice. Here, the exchange-breaking perturbation is $\delta t = 0.5t$.

## 6.2 Dice lattice and its projections

The Dice lattice [shown in the Fig. 15(a)] has the following Hamiltonian:

$$H_{\mathrm{Dc}} = -t \begin{pmatrix} 0 & 1 + e^{-i\vec{k}\cdot\vec{R}_1} + e^{-i\vec{k}\cdot\vec{R}_2} & 1 + e^{-i\vec{k}\cdot\vec{R}_1} + e^{i\vec{k}\cdot(\vec{R}_2-\vec{R}_1)} \\ 1 + e^{i\vec{k}\cdot\vec{R}_1} + e^{i\vec{k}\cdot\vec{R}_2} & 0 & 0 \\ 1 + e^{i\vec{k}\cdot\vec{R}_1} + e^{-i\vec{k}\cdot(\vec{R}_2-\vec{R}_1)} & 0 & 0 \end{pmatrix}, \quad (28)$$

where $\vec{R}_1 = (1,0)$ and $\vec{R}_2 = \left(\frac{1}{2}, \frac{\sqrt{3}}{2}\right)$. The two subsystems are the triangular lattice

$$H_{\mathrm{eff,T}}(E_0) = \frac{4t^2}{E_0}\left[\frac{3}{2} + \cos\left(\vec{k}\cdot\vec{R}_1\right) + \cos\left(\vec{k}\cdot\vec{R}_2\right) + \cos\left(\vec{k}\cdot(\vec{R}_1-\vec{R}_2)\right)\right], \quad (29)$$

and something which is a Graphene lattice with nnn and nnnn hoppings (which we may call the extended Graphene lattice):

$$H_{\mathrm{eff,xG}}(E_0) = \frac{t^2}{E_0}\begin{pmatrix} |h(\vec{k})|^2 & e^{-i\vec{k}\cdot\vec{R}_1}h^2(\vec{k}) \\ e^{i\vec{k}\cdot\vec{R}_1}h^{*2}(\vec{k}) & |h(\vec{k})|^2 \end{pmatrix}, \quad (30)$$

where $h(\vec{k}) = 1 + e^{i\vec{k}\cdot\vec{R}_1} + e^{i\vec{k}\cdot\vec{R}_2}$.

We can now define the following matrix

$$H_{TX} = \left(t_{AB}^1 + t_{AB}^2 e^{-i\vec{k}\cdot\vec{R}_1} + t_{AB}^3 e^{-i\vec{k}\cdot\vec{R}_2} \quad t_{AC}^1 + t_{AC}^2 e^{-i\vec{k}\cdot\vec{R}_1} + t_{AC}^3 e^{i\vec{k}\cdot(\vec{R}_2-\vec{R}_1)}\right). \quad (31)$$

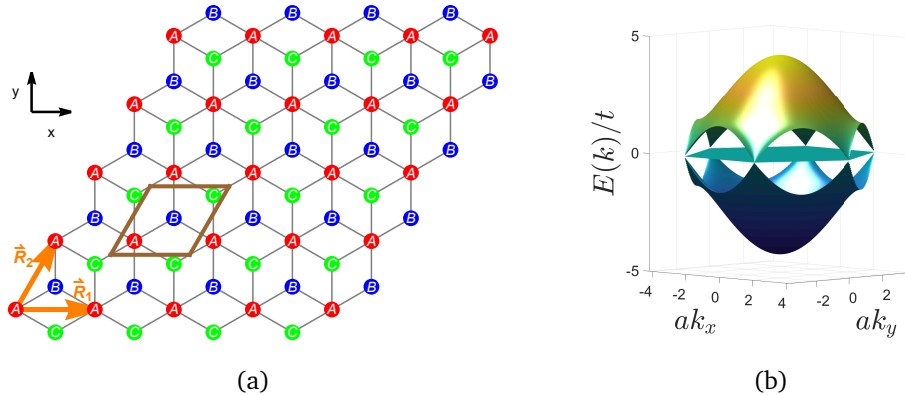

Figure 15: (a) Dice lattice with three atoms A, B, C in the unit cell and the translation vectors $\vec{R}_1$ and $\vec{R}_2$. (b) The respective energy spectrum with the flat band as well as particle-hole symmetry.

We can now break the path-exchange symmetry in the Dice lattice [see Fig. 17(b)] by selecting the parameterization $t_{AC}^1 = (t - \delta t)$, $t_{AC}^2 = (t - \delta t)$ and $t_{AC}^3 = (t + \delta t)$. We show in Fig. 17(d) the spectrum of the exchange-broken Dice lattice projected onto the extended Graphene lattice which shows the isolated flat band.

# 7 Beyond the nn approximation

Thus far we restricted ourselves to nn model of systems. In fact that was one of our motivations. In this section, we explore what happens if we consider small next nn (nnn) perturbations to the system in the form of $t_{nnn}$. This can be addressed by studying the following two classes. One class is where we include the nnn in the parent system, where we will lose the bipartite condition and hence also likely lose the flat band. The bandwidth of the "flat band" is determined by the ratio $t_{nnn}/t$. The other class is where we preserve the bipartite nature of the Hamiltonian but allow for next-nearest neighbor hopping only in the bipartite block. Since this preserves the bipartite nature, this also preserves the flat band, both in the parent system as well as the projected system.

## 7.1 Example 1: nnn in $H_5$

Consider the nnn hoppings in $H_5$ as described in Fig. 18 (a). The additional part to the Hamiltonian corresponding to the nnn hoppings is

$$H_5^{nnn} = t_{nnn} \begin{pmatrix} 0 & 0 & 0 & 0 & 0 \\ 0 & 0 & 0 & 0 & 0 \\ 0 & 0 & 0 & 1 + e^{i\vec{k}\cdot\vec{R}_2} & 1 + e^{i\vec{k}\cdot\vec{R}_1} \\ 0 & 0 & ... & 0 & 1 + e^{i\vec{k}\cdot(\vec{R}_1 - \vec{R}_2)} \\ 0 & 0 & c.c. & ... & 0 \end{pmatrix}. \tag{32}$$

This addition violates the bipartiteness condition for flat band and thus does not guarantee a flat band. The energy spectrum in Fig. 18 (b) demonstrates that the flat band ends up dispersing in this case.

We can still have higher order hoppings while preserving the flat band as long as the bipartite condition is satisfied. An example of such bipartite nnn hoppings are depicted in Fig.

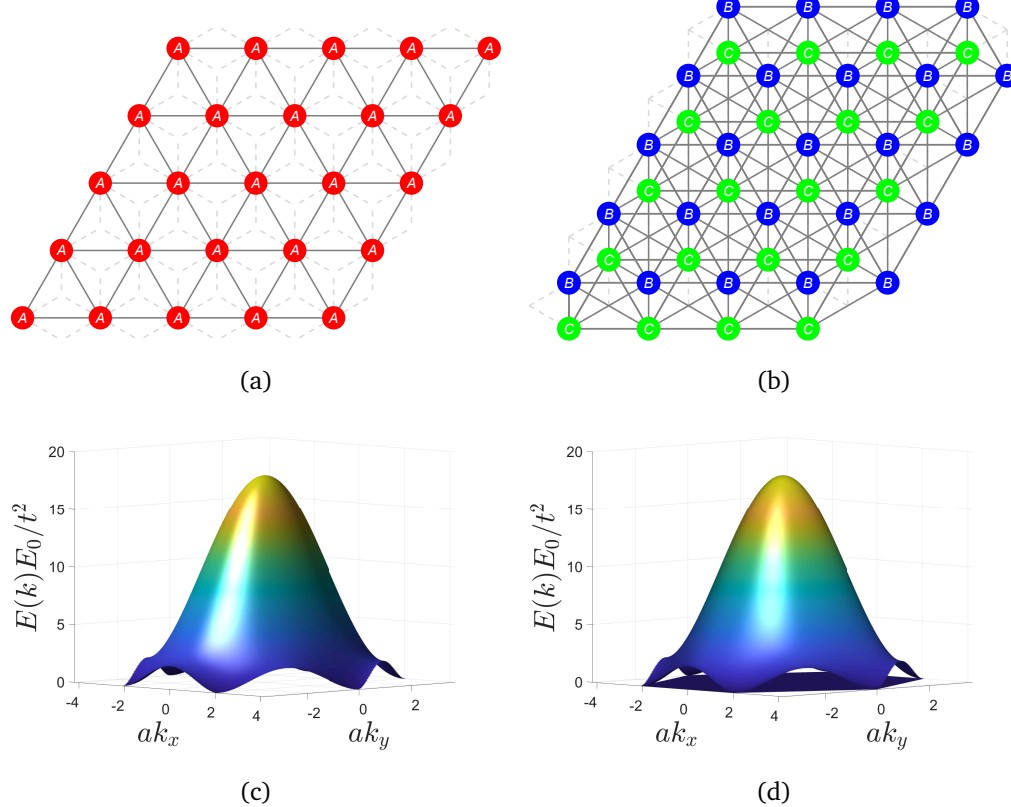

Figure 16: The two projected subsystems of the Dice lattice: (a) triangular lattice and (b) extended Graphene and their respective energy dispersions in (c) and (d). Note the flat band in the extended Graphene.

19(a). The correction to $H_5$ due to these hoppings is

$$
H_5^{bip-nnn} = t_{nnn}^{bip}
\begin{pmatrix}
0 & 0 & 1 + e^{i\vec{k}\cdot(\vec{R}_1-\vec{R}_2)} & e^{i\vec{k}\cdot(\vec{R}_2-\vec{R}_1)} + e^{i\vec{k}\cdot\vec{R}_2} & e^{i\vec{k}\cdot\vec{R}_1} + e^{i\vec{k}\cdot(\vec{R}_1-\vec{R}_2)} \\
0 & 0 & e^{i\vec{k}\cdot\vec{R}_2} + e^{i\vec{k}\cdot\vec{R}_1} & 1 + e^{-i\vec{k}\cdot\vec{R}_1} & e^{i\vec{k}\cdot(\vec{R}_2-\vec{R}_1)} + e^{-i\vec{k}\cdot\vec{R}_1} \\
\dots & \dots & 0 & 0 & 0 \\
\dots & \dots & 0 & 0 & 0 \\
c.c. & \dots & 0 & 0 & 0
\end{pmatrix} . \tag{33}
$$

The band structures for $H_5$ and the projected system are shown in Fig. 19 (b) and (c) to be preserving the flat band. In addition, breaking the path exchange symmetry in the bipartite-nnn $H_5$ lattice by modifying an arbitrary hopping lifts the degeneracy of the flat band. As an example, consider modifying the $H_{1,5}$ element from 1 to $1 + \Delta$. The respective band structure is shown in Fig. 19 (d).

## 7.2 Example 2: Lieb lattice

The same idea applies in the case of the Lieb lattice. Consider the true nnn hoppings in Fig. 20(a) and the bipartite nnn hoppings in (c). The additional part of the Hamiltonian that describes the true nnn hoppings is

$$
H_{Lb}^{nnn} = t_{nnn}
\begin{pmatrix}
0 & 0 & 0 \\
0 & 0 & 1 + e^{i\vec{k}\cdot\vec{R}_1} + e^{-i\vec{k}\cdot\vec{R}_2} + e^{i\vec{k}\cdot(\vec{R}_1-\vec{R}_2)} \\
0 & c.c. & 0
\end{pmatrix} . \tag{34}
$$

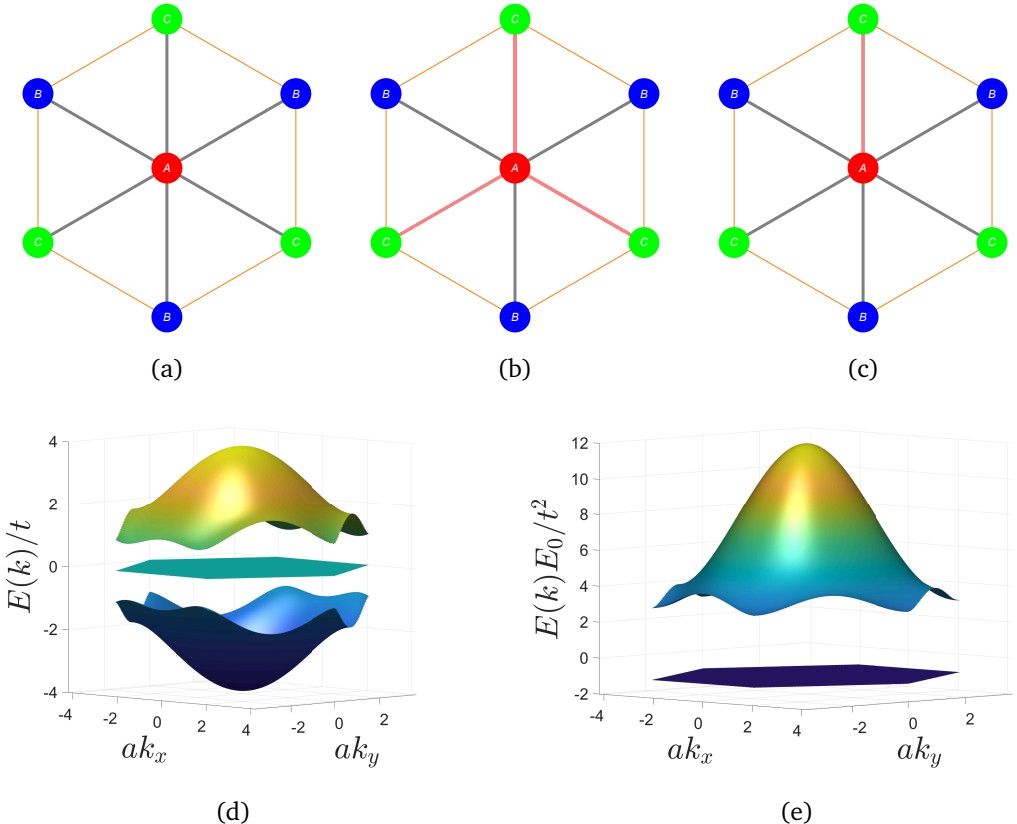

Figure 17: Hoppings in the unit cell of the Dice lattice. A is one subsystem and BC is the other. In the situations (a) and (b) the exchange-symmetry is preserved and so is the flat band degeneracy, whereas in (c) it is broken and the flat band degeneracy is lifted. (d) Isolated flat band in the spectrum of the exchange-broken Dice lattice and (e) the same but projected onto the extended Graphene lattice. Here, the exchange-breaking perturbation is $\delta t = t$.

For the bipartite nnn hoppings, the additional part is

$$
\begin{aligned}
&H_{Lb}^{bi-nnn} \\
&= t_{nnn}^{bip}
\begin{pmatrix}
0 & e^{i\vec{k}\cdot\vec{R}_2} + e^{-i\vec{k}\cdot\vec{R}_2} + e^{i\vec{k}\cdot(\vec{R}_2-\vec{R}_1)} + e^{-i\vec{k}\cdot(\vec{R}_1+\vec{R}_2)} & e^{i\vec{k}\cdot\vec{R}_1} + e^{-i\vec{k}\cdot\vec{R}_1} + e^{i\vec{k}\cdot(\vec{R}_1-\vec{R}_2)} + e^{-i\vec{k}\cdot(\vec{R}_1+\vec{R}_2)} \\
\dots & 0 & 0 \\
c.c. & 0 & 0
\end{pmatrix}.
\end{aligned}
\tag{35}
$$

Once again breaking the path exchange symmetry by modifying an arbitrary hopping in the Hamiltonian isolates the flat band in the bipartite-nnn Lieb lattice as shown in the band structure in Fig. 20 (f).

## 7.3 Flatness in path-exchange broken system with nnn hoppings

When true nnn hoppings are considered the dispersionless nature of the flat band is generally lost, but if the dispersion is sufficiently small, it may be thought of as a flat band. To determine the degree of flatness we compare the bandwidth of the flat band with the gap to its neighboring band. To begin, we fix a value for $\Delta$, which breaks the path exchange symmetry. Then, for varying the amplitude $t_{nnn}$ and plot the bandwidth of the flat band relative to its gap. Figure 21 shows that the isolated flat band (due to breaking path exchange symmetry) of

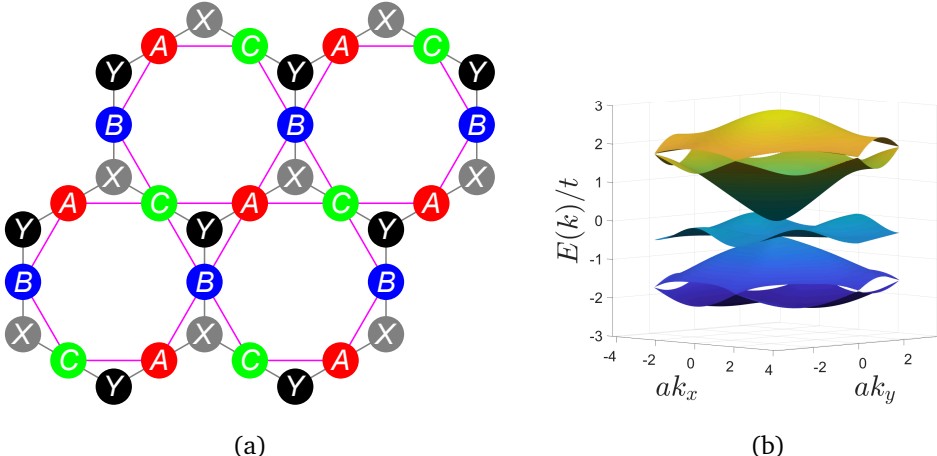

(a)              (b)

Figure 18: The nnn hoppings are shown in pink in (a). The corresponding energy spectrum is shown in (b). Note that the flat band becomes dispersive as the bipartite nature is lost due to nnn hoppings.

the $H_5$ system disperses as soon as nnn hoppings are considered and the bandwidth expands first linearly and then as a power law with stronger nnn hoppings.

## 8 Flat band with Chern-Simons flux distribution

In this section we demonstrate that the properties we outlined above also apply to systems with reduced translation symmetry such as in situations where the lattice is subject to a flux $\phi = 2\pi p/q$ per unit cell where p and q are co-primes. Such a system is usually modelled as a Hofstadter problem by attaching phases on the bonds such that when the pattern is extended to the entire lattice one finds a unit cell that is enlarged $q$-fold.

First, let us note a feature of the projection of the subsystems. Although we introduced the projection in the **k**-space, the same principles apply in the real space. This can be seen by performing a unitary transformation from the Bloch basis to tight-binding basis. It is advantageous to work in the real space as one would not have to worry about the unit-cell enlargement due to the flux attachment to the bonds. For definiteness, consider again the bipartite $H_5$ lattice with atoms $\{X_i, Y_i\}$ from one subsystem and $\{A_i, B_i, C_i\}$ in the other subsystem, where $i$ marks the unit cell. By a direct calculation it can be shown that if one were to project out the XY subsystem then the effective hopping that is induced in the ABC subsystem, say $t_{PQ}$ (where $P, Q \in \{A, B, C\}$), would be given by the sum of products of hoppings taking $P \rightarrow \{X, Y\} \rightarrow Q$. In the $H_5$ case every sum only consists of one term such that $t_{PQ} = t_{PX} t_{XQ}$ or $t_{PY} t_{YQ}$. This means that the phase associated with the effective bond $PQ$ would be the sum of phases along the path from $P$ to $Q$ in the original system.

Second, it is clear that since we started out with a bipartite system with different subsystem sizes, we would end up with a projected system with a flat band. However, it is interesting to observe the situation depicted in Fig. 22 where the parent system has phases $\phi_1$, $\phi_2$ and $\phi_3$ attached to the hoppings between the subsystems. Upon projection, using the result presented above, we find that $\phi_{AB} = \phi_1 + \phi_2$, $\phi_{BC} = \phi_3 - \phi_2$, and $\phi_{CA} = -\phi_3 - \phi_1$. This results in $\phi_{AB} + \phi_{BC} + \phi_{CA} = \oint \vec{A} \cdot d\vec{l} = 0$, i.e. the flux enclosed in the triangular regions is zero. This is in fact the scenario that is predicted to happen for in a Kagomé chiral spin-liquid which is modelled as fermions subject to a Chern-Simons (CS) field. Thus, the flux distribution that is implemented by the projection technique naturally accounts for the CS nature of the flux

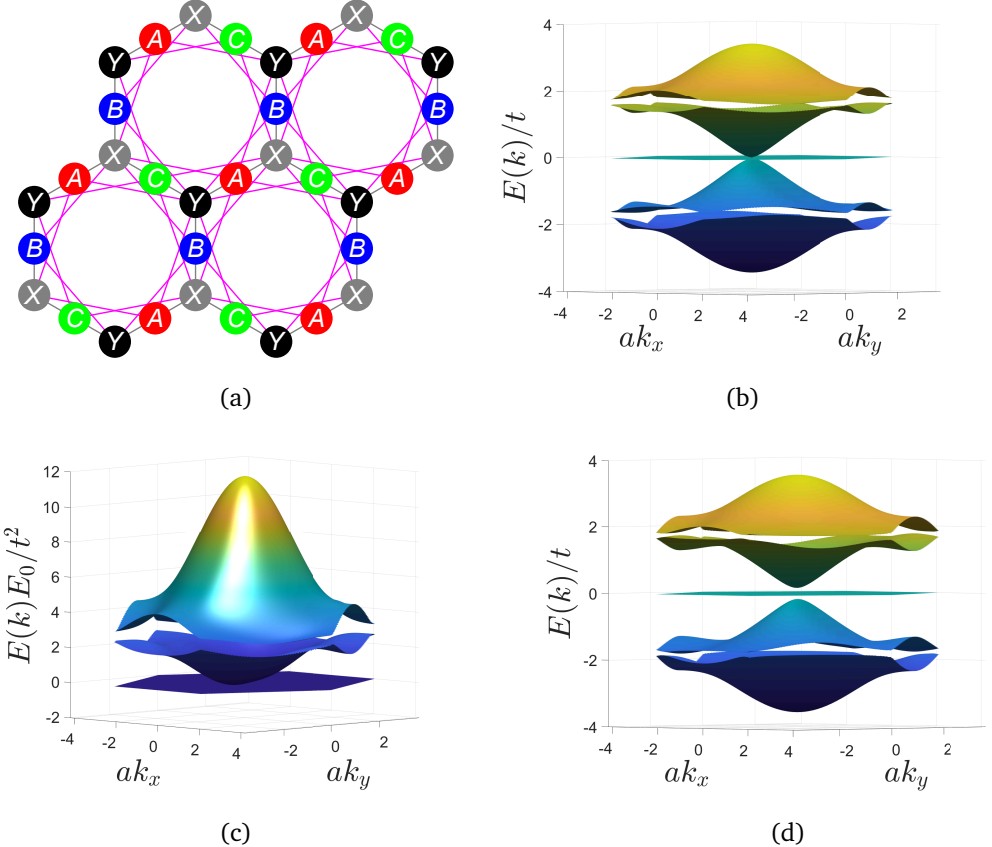

Figure 19: $H_5$ model with nnn hoppings in bipartite sense. The pink lines in (a) represent the nnn hoppings restricted to the different subsystems. The energy spectrum with $t_{nnn}^{bip} = 0.2t$ in (b) clearly shows that the flat band is preserved. The projected Kagomé sublattice inherits the flat band as shown in (c). When the path exchange symmetry is broken in the bipartite-nnn $H_5$ lattice, the flat band is isolated from the dispersive bands as shown in (d).

distribution and is probably a good tool to use to model systems with CS fields. It follows from this that in a Maxwell-like flux distribution (where the flux is proportional to the area enclosed, see Fig. 23), a flat band is not guaranteed. This is the reason why lattices with Maxwell-like fluxes have dispersive bands (in general), whereas those with CS-type fluxes can have flat bands. This comparison has been discussed in Fig. 4 of Ref. [28]. In that work, it was also pointed out that the flat band was gapped from the other dispersive bands. This is now easy to understand ( e.g., see Fig. 24) that the path-exchange symmetry is broken due to the phase attachments, which isolates the flat band.

As an explicit example let us consider the case with $\phi = \pi$. Here the unit cell is doubled ($q = 2$). The Hamiltonian is given by

$$H^{\phi=\pi} = \begin{pmatrix} 0 & H_{GK}^{\pi} \\ H_{KG}^{\pi} & 0 \end{pmatrix}, \tag{36}$$

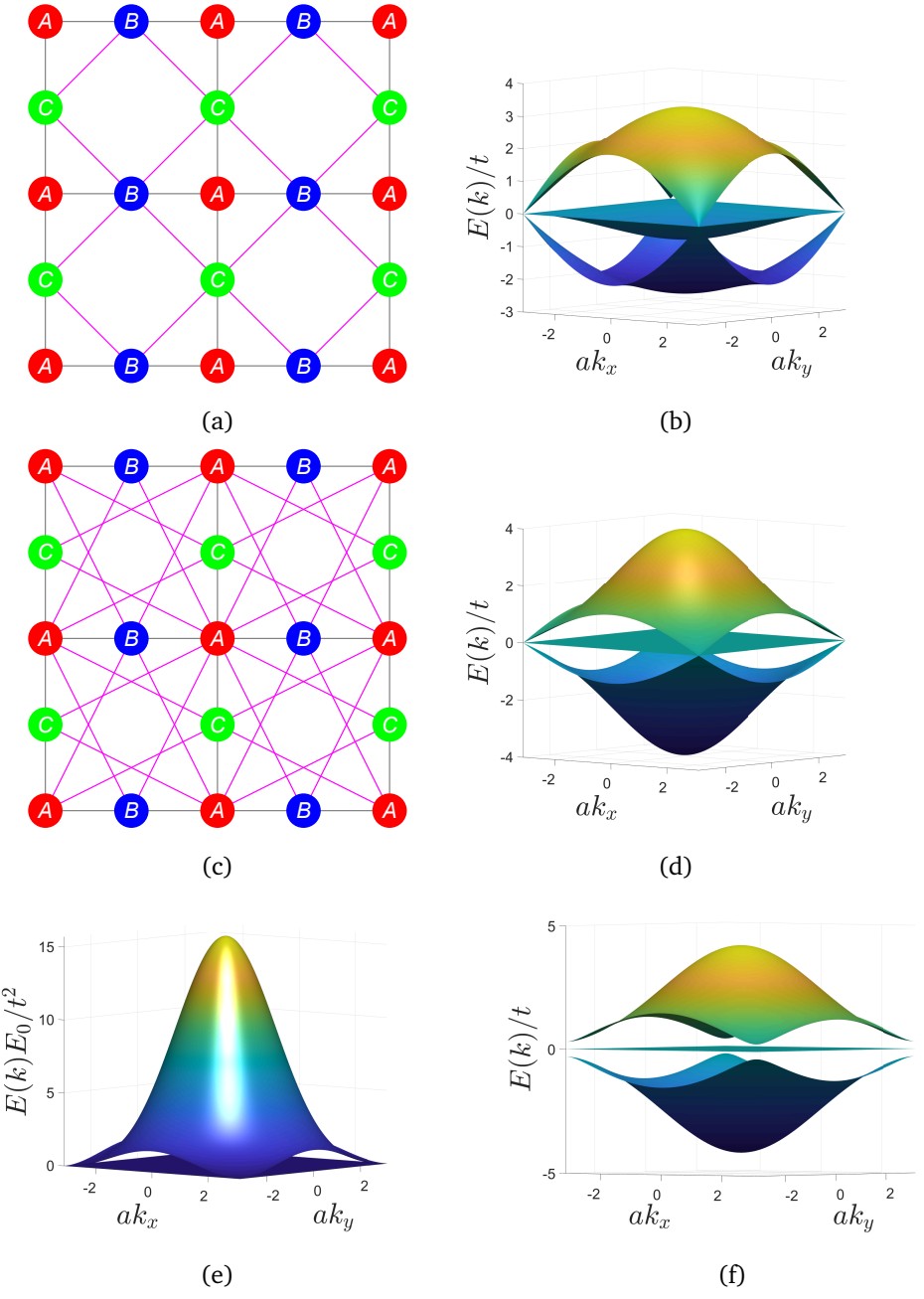

Figure 20: True nnn hoppings are colored in pink in (a). The energy spectrum with $t_{nnn} = 0.2t$ in (b) shows that the flat band in the middle is now dispersive. The nnn hoppings in the bipartite sense are shown in pink in (c). The spectrum with $t_{nnn}^{bip} = 0.2t$ in the parent system, (d), and the projected system, (e), show that the flat band is preserved. Breaking the path exchange symmetry in the bipartite-nnn Lieb results in isolating the flat band as shown in (f).

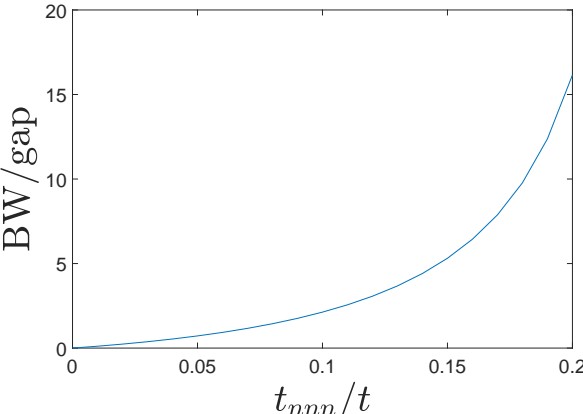

Figure 21: The evolution of the bandwidth of the flat band in path exchange broken $H_5$ (with $\Delta = 0.2$) with varying $t_{nnn}$. Increasing strength of the nnn hoppings expands the bandwidth of the flat band.

where

$$
H_{KG}^{\pi} = -t
\begin{pmatrix}
1 & 0 & 0 & e^{-i\vec{k}\cdot\vec{R}_2} \\
0 & e^{-i\pi} & e^{-i\vec{k}\cdot\vec{R}_2} & 0 \\
1 & 0 & 1 & 0 \\
0 & 1 & 0 & 1 \\
1 & 0 & e^{i\vec{k}\cdot(\vec{R}_1-\vec{R}_2)} & 0 \\
0 & e^{-i\pi} & 0 & e^{i\vec{k}\cdot(\vec{R}_1-\vec{R}_2)}
\end{pmatrix},
\tag{37}
$$

and $R_1 = (1,0)$ and $R_2 = \left(\frac{1}{2}, \frac{\sqrt{3}}{2}\right)$. The phase attachments are shown in Fig. 24 and the band structure for the $H_5$ system and the projected systems are shown in Fig. 25, which clearly show the isolated flat band. This feature remains for all values of $\phi \in (0, 2\pi)$.

## 9 Conclusion

Although there exist many works in the literature on designing systems with flat bands, they suffer from one or many of the following shortcomings: the need for long-ranged hoppings; the need for fine-tuning of parameters even with nn approximation; the need for staggered fluxes. While these are all valid techniques, they could be seen as shortcomings because implementing long-range hoppings is often a design challenge. So is fine-tuning parameters of the Hamiltonian. Breaking a discrete symmetry such as the time-reversal symmetry is certainly viable, however, generating staggered flux may not always be a straightforward task. In this work, we proposed a straightforward design method based on the bipartiteness of a system (which is not strictly necessary) and Löwdin's projection to generate flat band systems. In this prescription, we were able to identify that if there exists a path-exchange symmetry in the bipartite system, with different sizes of the subsystems, the flat band would appear degenerate with other dispersive bands. And by breaking the symmetry, it is possible to isolate the flat band. We then showed that projecting out various subsystems from this parent bipartite system still leaves behind a subsystem with an isolated flat band. We demonstrated the verity of all the above points of the prescription by applying it to the $H_5$ lattice (from which we can project out the hexagonal lattice and the Kagomé lattice: the latter having the spectrum of the former plus a flat band), Lieb and Dice lattices (from which we can project out the square and

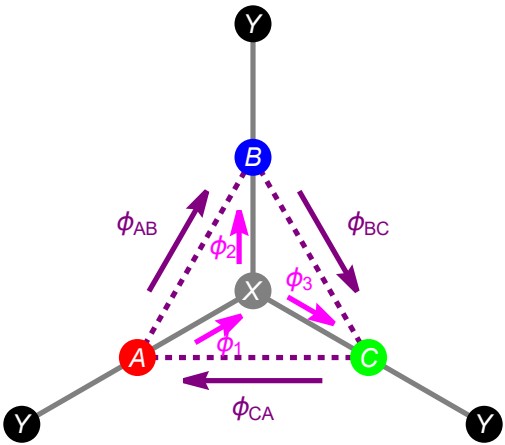

Figure 22: The hoppings inside the unit cell of the $H_5$ lattice, but with phases $\phi_{1,2,3}$ attached as shown. Upon projecting out the XY subsystem, the effective hoppings (dashed lines) acquire phases $\phi_{AB,BC,CA}$.

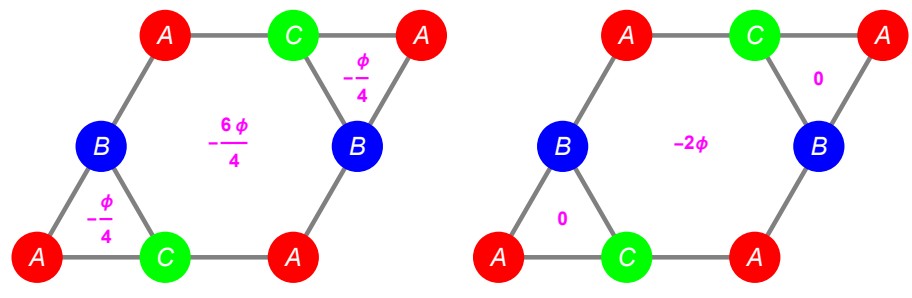

(a) Maxwell type flux distribution     (b) Chern-Simons type flux distribution

Figure 23: Under Maxwell type field, the flux is proportional to area whereas in the CS case, the flux is concentrated in one region of the unit cell, with the triangular parts having zero flux.

triangular lattices, respectively, and special lattices that share their respective spectra plus a flat band). We also showed that relaxing the bipartite condition in the subsystem that is being projected out still maintains the flat band. In this sense, we only need to couple a non-hopping (or weakly hopping) system to a background system, which is smaller in size, to get a flat band.

Apart from this main result, we even developed a special parameterization of the Kagomé lattice to show that it is possible to devise flat band conditions, different from what is already in the literature and showed that this construction is automatically implemented if arrived at from the parent bipartite system. We even showed that starting from the parent bipartite system, we can reproduce the conditions for the existence and isolation of flat bands that were discussed previously in the literature (about staggered $\pi$ fluxes [44] and inversion broken Kagomé [51]). Finally, we demonstrated that the existence of a flat band in the chiral spin-liquid state of the Kagomé lattice that originates due to coupling to a Chern Simons like field is also consistent with the rules we identify in this work. This also suggests that the Chern Simons type flux in lattices could be understood in terms of Maxwellian flux coupled to bipartite systems.

We also believe that the prescription suggested here should be practically implementable in photonic-crystal lattices or even in cavity QED techniques such as in Refs. [57,58]. It is also worth noting that since our statements about isolating flat bands are essentially derived from

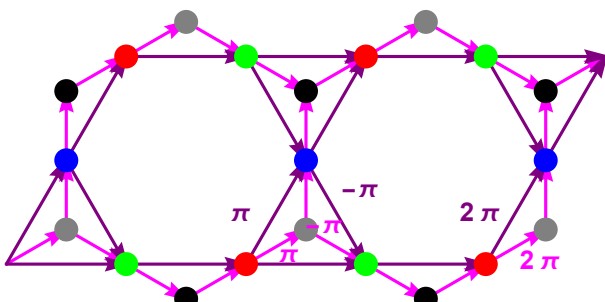

Figure 24: Chern-Simons flux attachment prescription for $\phi = \pi$. All the flux is concentrated on the middle hexagon and the flux through the triangular regions is zero. The unit cell is doubled. The purple lines indicate the hoppings in the projected subsystem and each parallelogram formed by these lines is the original unit cell. The pink lines represent the hoppings of the parent system. The arrows indicate the phases, which are all zero except the three in purple and pink each with $(\pi, -\pi, 2\pi)$.

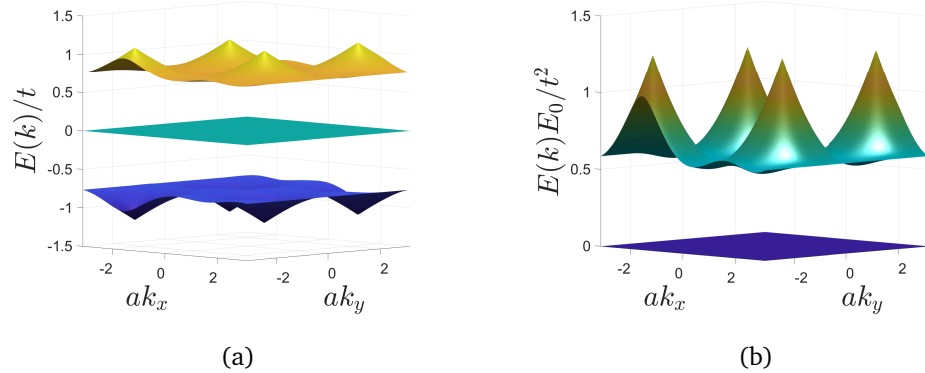

Figure 25: (a) Energy spectrum of $H_5$ with CS field with flux $\phi = \pi$ (b) Energy spectrum of $H_{\text{eff},K}$. In both cases notice the flat band at $E = 0$. We are only showing the bands closest to the flat band.

properties of bipartite graphs, they can be extended to lattices in hyperbolic spaces where it is easier to accommodate many neighbors at the same distance than in euclidean space. Since a Bloch theorem exists also in the hyperbolic space [69], we expect our formalism to apply readily to such systems at each hyperbolic-$\vec{k}$ point in the hyperbolic Brillouin zone. Isolating the various subsystems can be done, e.g. simply by introducing appropriate onsite energies to split the subsystems in energy, and then tuning the energy scale of any probe to be near the respective onsite energies of interest. This will a subject for a future work.

Finally, having described our formalism and the physical intuition behind it, it is fitting to conclude by drawing some connections to the methods of other flat band studies cited earlier. The Gram matrix technique cited earlier [52] also relies on the $M^\dagger M$ construction we introduced, however, we provide a physical way to arrive at such matrices from parent systems that do not require fine-tuning whereas in that work it was just posed as a mathematical problem leading to fine-tuning of the Hamiltonian. We also note that recently there was a notion of singular flat band introduced into the literature [70, 71] wherein it was stated that such bands could not be isolated without dispersing. While the classification is useful, the work does not outline a procedure to find the singular or non-singular flatband. As is evident, our work clearly outlines that the perturbations to the parent system, of any spatial dimension

and any Hilbert-space dimension, that breaks the path-exchange symmetry results in a system with non-singular flat bands.

There are several interesting avenues to pursue using this construction. It may be possible to study the role of electronic correlations in Kagomé and Graphene by studying them in the parent $H_5$ system. Given that we have a systematic way to isolate flat bands, it can serve as a test bench to investigate fractional Quantum Hall state formation in the absence of a magnetic field. Further, it was stated in Ref. [43] that the band touching was protected by topology in the Kagomé system. However, we have seen that the band touching is protected by a path exchange symmetry in a parent bipartite system. This could suggest a connection between topology and exchange-symmetries in a higher-dimensional Hilbert space. It will also be interesting to explore the link between the path-exchange symmetry and the Shubnikov groups as elaborated on in Ref. [55].

## Acknowledgments

The authors would like to acknowledge useful exchanges with A. Andreanov.

**Funding information** This work was funded by the Natural Sciences and Engineering Research Council of Canada (NSERC) Grant No. RGPIN-2019-05486 (S.M.).

## A Useful relations between matrix elements $\tilde{\alpha}_i$

There are a couple of identities involving the matrix elements of the Hamiltonian that can be derived straightforwardly. For the parameter $|r| < 1$, we observe that

$$|\tilde{\alpha}_i|^2 = 2e^{-h}\left[\cosh h + \cos\left(\vec{k}\cdot\vec{R}_i\right)\right],$$

and the quantity $A = 2\text{Re}[\tilde{\alpha}_1^*\tilde{\alpha}_2\tilde{\alpha}_3^*]$ evaluates to

$$A = 2\left[1 + \left(e^{-h} + e^{-2h}\right)\sum_i \cos\left(\vec{k}\cdot\vec{R}_i\right) + e^{-3h}\right].$$

Using these we arrive at the relation

$$\sum_i |\tilde{\alpha}_i|^2 = 2e^{-h}\left[3\cosh h + \sum_i \cos\left(\vec{k}\cdot\vec{R}_i\right)\right].$$

Observing that both $\sum_i |\tilde{\alpha}_i|^2$ and $A$ have $\sum_i \cos(\vec{k}\cdot\vec{R}_i)$ in them as the only $\vec{k}$-dependent term, it is possible to search for $f$ that satisfies the flat band condition in Eq. (5) by equating their coefficients, which leads to $f = 2e^{-\frac{h}{2}}\cosh\frac{h}{2}$.

If $|r| > 1$, then the identification $\frac{1-r}{1+r} = e^{-h}$ can still be made if $h$ is extended to the complex plane. In particular, we note that as $|r| > 1$, $h$ acquires a step jump in the imaginary part from 0 to $\pm i\pi$. The sign is ambiguous but this does not affect our analysis and we shall stick to the choice of $i\pi$. That is, $h \to h_c$ such that $e^{-\text{Re}[h]} = \frac{r-1}{r+1}$. This extension to the complex plane also results in $[e^{-h_c}]^* = e^{-h_c}$, thus acting like a real number. This ensures that all the above identities above are still valid with $h \to h_c$. In that case, the resulting parameter $f$ is given by $f = 2e^{-\frac{h_c}{2}}\cosh\frac{h_c}{2} = 2e^{-\frac{\text{Re}h_c}{2}}\sinh\frac{\text{Re}h_c}{2}$, where $\frac{1-r}{1+r} = e^{-\text{Re}h_c}$.

# B  Properties of non-square matrices

Consider a non-square matrix $M_{n \times m}$ with $m > n$ for definiteness. Two square matrices could be constructed from this: $[MM^\dagger]_{n \times n}$ and $[M^\dagger M]_{m \times m}$. Now construct the square matrix $\tilde{M}_{m \times m}$ by padding $m - n$ rows of zero to $M$. Then we have

$$\tilde{M}\tilde{M}^\dagger = \begin{pmatrix} [MM^\dagger]_{n \times n} & 0_{n \times (n-m)} \\ 0_{n \times (n-m)} & 0_{(n-m) \times (n-m)} \end{pmatrix},$$

and

$$\tilde{M}^\dagger \tilde{M} = M^\dagger M.$$

From properties of matrices we have $\text{Det}[\tilde{M}\tilde{M}^\dagger] = \text{Det}[\tilde{M}^\dagger \tilde{M}]$ which is the product of their eigenvalues. Since $\tilde{M}\tilde{M}^\dagger$ explicitly has $m - n$ 0's as eigenvalues, and that this construction is possible for any $M$, it follows that $\tilde{M}^\dagger \tilde{M}$ and hence $M^\dagger M$ must have (i) the same number of 0 eigenvalues; (ii) the same non-zero eigenvalues.

In other words, what we have argued is that for a non-square matrix $M_{n \times m}$, $MM^\dagger$ and $M^\dagger M$ have the same eigenvalues with one of them having $|n - m|$ zeros as additional eigenvalues.

# C  Identifying the path-exchange symmetry

Consider a generic flat band system generated based on a bipartite system whose subsystems have size $n \times n$ and $m \times m$ with $n < m$. In such a system, there are $m - n$ flat bands at $E = 0$. This means that there are $m - n$ additional redundant equations (rows) in the system for any $\mathbf{k}$. If the flat bands are additionally degenerate with any of the other dispersive bands at a particular $\mathbf{k}$-point, then at that $\mathbf{k}$-point there must by additional equations (rows) that are redundant. A redundant equation (row) can be reduced to a scaled version of the other linearly independent equations (rows). For generic scenarios in nn bipartite systems, it is sufficient to just look at the scaling property. In a bipartite system of the form

$$\hat{H} = \begin{pmatrix} 0 & \hat{h}^\dagger_{n \times m} \\ \hat{h}_{m \times n} & 0 \end{pmatrix}, \tag{C.1}$$

all the information is contained in the block $\hat{h}_{m \times n}$, which has $m$ equations and $n$ unknowns. If such a system has $E = 0$ solutions, then only $n$ of these are linearly independent. If one identifies $z$ rows that could be scaled (we call them $z$ zero rows), then linear algebra dictates that there would be $d = 2(z - [m - n]) + 1$ degeneracies in the system. That is, for every new row, we introduce two degeneracies. The factor of 2 arises from the fact that each time we reduce a row in $\hat{h}$, we reduce one in $\hat{h}^\dagger$ (this won't be the case for non-bipartite cases as will be discussed shortly). To extract the physical meaning behind this, note that each redundant row, if not trivially zero, is a scaled version of the linearly independent rows. Since a given row in the $\hat{h}$ block contains the hopping to an atom, say $A$, of the larger subsystem from all the atoms of the smaller subsystem, the redundancy of another row would imply that the hopping to another atom, say $B$, of the larger subsystem is a scaled version of $A$. That is, the set $\{X_i A\}$ and set $\{X_i B\}$ (where $\{X_i\}$ is the set of atoms in the smaller subsystem) are related via $\{X_i A\} = \alpha \{X_i B\}$. If $X_i A$ and $X_i B$ are not trivial, then we can construct the set of ratios $\{\frac{X_i A}{X_i B}\}$. A redundancy of a row requires the set $\{\frac{X_i A}{X_i B}\}$ to be a unit set and we say that the paths to $A$ and to $B$ are exchangeable. In simpler terms, for a bipartite system, for $z$ unique path-exchanges we would have $2(z - [m - n]) + 1$ degeneracies. Thus, breaking/reducing the number of path-exchanges would lift/reduce the degeneracies. If we now consider the system

$$\hat{H} = \begin{pmatrix} \hat{s}_{n \times n} & \hat{h}^\dagger_{n \times m} \\ \hat{h}_{m \times n} & 0 \end{pmatrix}, \tag{C.2}$$

where $\hat{s}$ is non-trivial, then for $z$ path-exchanges, we would have $d = (z - [m-n]) + 1$ degeneracies. We lose the factor of 2 due to presence of $\hat{s}$.

As an example, consider the bipartite $H_5$ lattice with $m = 3, n = 2$. Here the formula that applies is $d = 2(z-1) + 1 = 2z - 1$. The number of reducible rows ($z$) can be 1 or 2. Thus, we can have a degeneracy $d$ of 1 (non degenerate) or 3 (triply degenerate) at $E = 0$ or none. So, if the $\hat{h}$ block looks like

$$\begin{pmatrix} t_1 & t_2 \\ t_3 & t_4 e^{i\vec{k}\cdot\vec{R}_1} \\ t_5 & t_6 e^{i\vec{k}\cdot\vec{R}_2} \end{pmatrix}, \tag{C.3}$$

the case with two path-exchanges would look like

$$\frac{t_3}{t_1} = \frac{t_4 e^{i\vec{k}\cdot\vec{R}_1}}{t_2}, \quad \text{and} \quad \frac{t_6 e^{i\vec{k}\cdot\vec{R}_2}}{t_2} = \frac{t_5}{t_1}, \tag{C.4}$$

then $d = 2(2 - [m-n]) + 1 = 3$, a triple degeneracy. Because the hoppings are real, this condition is only satisfied at the $\Gamma$-point ($\mathbf{k} = 0$). Breaking any of the path-exchange reduces the degeneracy by 2 and hence removing the degeneracy in this case.

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
