# Peer review of "Isolated flat bands in 2D lattices based on a novel path-exchange symmetry"

_SciPost Physics, doi:SciPost Phys. 15, 139 (2023)_

## Round 1 · Referee Report · Philippe St-Jean (Referee 1) · 2023-5-18

Strengths

1- Impact: the work is a very interesting perspective on a general method for engineering flat energy bands. There have been many proposals and demonstrations for implementing crystalline structures that could lead to such flat bands but very few have presented generic methods for realizing them. For example, Ref. [52] from the Bernevig group in 2022, is a recent important example, and the current manuscript provides an interesting alternative approach. Although one might find important similarities between these two works, the two approaches are sufficiently distinct to gain intuition and/or insightful perspectives.

2- Relevance: the subject of flat bands is a timely and important topic. A lot of effort is currently devoted to understand the emergence of these dispersive-less bands, notably because the interaction energy dominates and is expected to give rise to interesting many-body excitations.

2- Clarity: the paper is very well-written and allows for a clear understanding of the benefit of the method developed.

Weaknesses

1- I find that the paper lacks quantitative descriptions of its main results. For example, it is hard to appreciate directly the flatness of the bands obtained as only figures of band structures are presented. I would have appreciated to see figures presenting directly the flatness of the bands as a function of the relevant parameters.

2- The paper also lacks discussion on the limitations of the model developed. The entire work is limited to systems that are well described in the tight-binding approximation. However, this is only an approximation and real systems will deviate from it; in that context, what would be the influence of these deviations from tight-binding?

Report

This manuscript reports a theoretical work where a generic model for engineering flat bands in crystalline systems (solid-state, photonics, atomic...) is presented. I think this is a timely and and important topic, hence this paper provides insightful and interesting perspectives on this field. I think this manuscript is worth publishing in SciPost, provided that the authors address the minor points I raised above.

Requested changes

1- Include dedicated figures presenting a quantitative metric describing the flatness of the bands (e.g. E_max - E_min for a band as a function of the relevant parameters like r). I would also like to see figures presenting the evolution of the gap energy as a function of these parameters to better see the lift of degeneracy between the flat and dispersive bands.

2- Include a discussion (even a brief one) on the impact of deviating from the tight-binding approximation. This could be done, e.g., by including small next-nearest-neighbor couplings or other orbitals, and discussing how this impact the flatness and the gaps. This will be crucial to better understand the robustness of their scheme.

  • validity: high
  • significance: good
  • originality: good
  • clarity: high
  • formatting: excellent
  • grammar: excellent

Author:  Saurabh Maiti  on 2023-07-14  [id 3813]

(in reply to Report 1 by Philippe St-Jean on 2023-05-18)
Category:
remark

Thank you for your time and evaluation of our work. You can see our detailed response in the resubmission letter. I mostly wanted to address your concern about including the next-nearest-neighbor(nnn) terms. We think you are completely justified in wondering what happens when these imperfections are included. To address this we included a section in the resubmitted version. But I just wanted to emphasize two points: 1. The formulation is exact. But to future readers I wanted to clarify that your question has to do not with the formalism but the applicability of it to real systems where there are nnn effects (which is a very relevant point). 2. The formalism still handles nnn well if the bipartite conditions is maintained. But in space respecting Euclidian geometry, the type of nnn usually breaks the bipartite nature. And this is harmful: it disperses rather quickly. We quantify this in a plot. We also quantify the gaps of the isolated flat bands in parent systems and projected systems. The effect size in projected systems usually gets smaller.

---

## Round 1 · Referee Report · Alexei Andreanov (Referee 2) · 2023-5-31

Strengths

1- Impact: classification and construction of flatbands is an important problem and any new approached are more than welcome. Even more important is establishing a connection between flatbands and symmetries.
2- Relevance: flat bands is an active topic of research motivated by their sensitivity to perturbations, in particular interactions which makes flatband models natural testbeds for novel and exotic phases of matter.

Weaknesses

1- I think the presentation of this symmetry could be further improved. For instance in Fig. 10 it is not immediately obvious why the symmetry is present in (c) but not in (d)?

Report

The authors propose a novel method to construct tight-binding Hamiltonians with a flat band -- dispersionless energy band, and derive a criterion for the flat band to be gapped away from the other (dispersive bands). The construction proceeds through projecting out the minority subsystem of the parent chiral sytem. The mismatch between the sizes of the majority and minority sublattices and the resulting rectangular matrix guarantees the presence of the flatband. This argument persists also in cases where the chiral symmetry is partially broken and hoppings within the minority subsystem are allowed. Whether the flatband is gapped or gappless is determined by the path-exchange symmetry related to how the minority and majority sublattices are connected.

This is an interesting and important work on flatbands that is definitely worth being published at SciPost Physics.

Requested changes

1- The deformation of the kagome lattice considered in Sec. 3 is known as breathing lattices in frustrated magnetism, and could be a convenient name for the deformation.
2- Within the Loedwin method flatband is always the GS (or the highest state in the spectrum) since effective Hamiltonian is positive definite. Is there a possibility to adapt/modify the method to flatbands in the middle of the spectrum?
3- Is there any connection to the following work: PHYSICAL REVIEW A 102, 053305 (2020) - Building flat-band lattice models from Gram matrices?
4- Does the Loedwin projection method extend to the case of other symmetries, which enforce the E -> -E symmetry of the spectrum, like (anti-)PT, etc?
5- The authors illustrate their findings with the Lieb and dice lattices which both have chiral symmetry. I wonder if the authors tried to take any Hamiltonian with a flatband as a ground state and reconstruct the parent lattice? The simplest example perhaps would be the sawtooth chain. Naively that seems to be possible, since the Hamiltonian can be made positive definite, and can be decomposed into a "square" of a rectangular matrix. However the hoppings might not be short-range again. If the answer is positive and there is a way to reconstruct the parent lattices, this would imply the hidden symmetry behind at least a subclass of flatbands.
6- Sec. 4.1: going beyond bipartiteness -- does this preserve short range hopping in general? or the long-range hoppings could be generated in this case?
7- It is important to compare/relate your path-exchange symmetry mechanism for the band touchings to the previously developed criterion for critical flat bands, e.g. flat bands with a touching: see "Singular flat bands", W Rhim, BJ Yang - Advances in Physics: X, 2021 - Taylor & Francis.

  • validity: top
  • significance: high
  • originality: good
  • clarity: high
  • formatting: excellent
  • grammar: excellent

Author:  Saurabh Maiti  on 2023-07-14  [id 3812]

(in reply to Report 2 by Alexei Andreanov on 2023-05-31)
Category:
remark

Thank you for your evaluation and the many useful suggestion. You can see our detailed response in the resubmission letter. I mostly wanted to touch on suggestions #4 and #5 which are relevant "next-step" questions and won't be addressed directly in our manuscript.
#4: This is a very relevant direction to think along. A good way to rephrase your question is to ask how do these symmetries/conditions emerge in the projected system from the parent system. Do they correspond to something simple? I hop you will agree with this that this question merits its own consideration separate from this work.
#5: We suspect that the parent system may not be unique, in general. We can construct two subsystems to connect to the subsystem of our interest and project both of them out. This is what stopped us from attempting this inverse problem. It certainly is an interesting idea to explore and again we hope that not including this in the current manuscript is reasonable.

---

## Round 2 · Referee Report · Philippe St-Jean (Referee 1) · 2023-7-18

Report

The authors have well addressed my comments. I now recommend the publication of their manuscript.

Requested changes

The authors have well addressed my comments. I now recommend their manuscript for publication.

---

## Round 2 · Referee Report · Alexei Andreanov (Referee 2) · 2023-8-1

Report

I am happy to see that the authors addressed the comments and remarks by the reviewers and gladly recommend the manuscript for publication.

---

## Round 2 · Author Response

We thank the referees for careful review of our manuscript and their positive and very supportive evaluation of it. We are happy to address the questions and the changes requested by them. Below is a detailed response to each referee’s requests. We are additionally going to upload this response to each referee's report for their convenience.

Addressing comments from Referee 1 We thank the referee for recognizing the relevance and message of this work, its connection and independence from Ref 52 [now 54], and also for the suggestions below which will prove very helpful to the readers.

  • Potential weaknesses "I find that the paper lacks quantitative descriptions of its main results. For example, it is hard to appreciate directly the flatness of the bands obtained as only figures of band structures are presented. I would have appreciated to see figures presenting directly the flatness of the bands as a function of the relevant parameters" "The paper also lacks discussion on the limitations of the model developed. The entire work is limited to systems that are well described in the tight-binding approximation. However, this is only an approximation and real systems will deviate from it; in that context, what would be the influence of these deviations from tight-binding?"

Response: Both points have been aptly pointed out. But the weaknesses identified are, in part, due to the scope of the work which is aimed at establishing the theory behind the claims made. In particular, regarding the flatness of the band, we adhered to an ideal standard (which is stricter than the referee’s ask) where we only claim something to be a flat band if it is exactly flat. That is the reason behind zooming in on some plots to emphasize this. In other words, any deviation whatsoever immediately disqualifies us from calling the band flat. We understand that experimentally there is always a tolerance and many non-flat bands can fall under the category of flat bands, but this would be a separate issue. The second point, however is very relevant and we added a section to address this exclusively (see below).

Requested changes "Include dedicated figures presenting a quantitative metric describing the flatness of the bands. I would also like to see figures presenting the evolution of the gap energy as a function of these parameters to better see the lift of degeneracy between the flat and dispersive bands "Include a discussion (even a brief one) on the impact of deviating from the tight-binding approximation. This could be done, e.g., by including small next-nearest-neighbor couplings or other orbitals, and discussing how this impact the flatness and the gaps. This will be crucial to better understand the robustness of their scheme.

Response: We added a subsection 5.1 discussing the evolution of the gap with the symmetry-breaking parameter (along with figures 11a and 11b). It scales linearly with this parameter in the parent system and quadratically in the projected system. We thank the referee for suggesting this very necessary inclusion. As stated above, we didn’t find it fitting to include the evolution of bandwidth of the flat band as they remain zero for all our cases. However, we did include one (Fig. 21) which shows the evolution of the bandwidth of the "flat band" when the nnn hoppings are included in the parent system (which results in the loss of the flatness). We added section 7 titled ‘Beyond the nn approximation’ where we present cases where our flatband are preserved and cases when they are lost (also quantifying the loss by measuring the bandwidth). The result is that if we preserve the bipartite structure, any order of the hoppings will still preserve the flat band in the parent and the projected system. The resulting Hamiltonian, although allowed, would be unrealistic. In realistic nnn cases, we lose the flatband. But this is not of concern to the theory which presents a new perspective on thinking about origin of flat bands. We believe that by implementing the above changes, we have faithfully addressed both the referee requests.

Addressing comments from Referee 2 We thank the referee for stating the importance of this work and reinstating the need to study flat bands more.

Potential weaknesses "I think the presentation of this symmetry could be further improved. For instance in Fig. 10 it is not immediately obvious why the symmetry is present in (c) but not in (d)

Response: We thank the referee for bringing this up. We clarified the test around the discussion of the symmetry and also included examples of applying the stated formula in relation to figure 10 (lines 417-444). To answer the referee’s Q: What happens in (c) is that the ratio of going from X to B and X to C is the same as the ratio of going from Y to B and Y to C. The entire path X-B-Y and X-C-Y is a scaled version of each other and hence identical. The same applies to X-A-Y and X-B-Y; and also X-A-Y and X-C-Y. But in (d), X-B-Y and X-C-Y is no longer scaled version and neither are X-A-Y and X-C-Y. But X-A-Y and X-B-Y are and hence there is one path exchange left. This gets down z=2 to z=1. Mathematically, z is simply the number of reducible rows, the path-exchange idea is an interpretation.

Requested changes [---denotes our response] • The deformation of the Kagome lattice considered in Sec. 3 is known as breathing lattices in frustrated magnetism, and could be a convenient name for the deformation. ---Yes, this was included already under “Physical interpretation of r-parameter section”. • Within the Loedwin method flatband is always the GS (or the highest state in the spectrum) since effective Hamiltonian is positive definite. Is there a possibility to adapt/modify the method to flatbands in the middle of the spectrum? ---The short answer is that the eigenvalues of the projected system are square of the parent. This is what ensures the the flat band is at zero while the others are above it (as the referee already noted). There could be additional flat bands in the parent system that can still end up in the middle of the bands after projection. If there is chiral symmetry, then after projection both subsystems will have the in-between- bands flat bands, but the extremity flat band will always tag along with the larger subsystem. The explanation for why the flatband is always at the extremity is now provided in lines 368-375. • Is there any connection to the following work: PHYSICAL REVIEW A 102, 053305(2020) - Building flat-band lattice models from Gram matrices? ---Yes, there is and our method clearly arrives at that condition upon performing projection. When a Hamiltonian can be written as H = T †T , where T is a linear transformation from V to V ′, it can be interpreted a Gram matrix which guarantees a flat band. Our bipartite construction, after the projection precisely produces Gram matrices, guaranteeing flat band. Upon the referee’s suggestion, we included a paragraph (lines 664-675) in the conclusion presenting the links to previous works on the construction of flat band systems. We feel the conclusion is the best place to do it as by this time the reader will have a good idea of our method. • Does the Loedwin projection method extend to the case of other symmetries, which enforce the E → -E symmetry of the spectrum, like (anti-)PT, etc? ---The question, on one hand, is a bit disconnected to the main goal of the work: to figure out a non-fine-tuned way to generate flat bands. On the other hand, it may be seen as connected as the stated symmetries do enforce flat bands in some systems. Whether Lowdin’s projection preserves this, or arrives at it will have to be investigated in some detail to make a general claim. But we can say with surety, that for systems where the stated symmetries induce a flat band, those are naturally covered (arrived at) in our formalism as our statements are really at the level of bipartite graphs and thus must encompass every sub-case that follows. But exploring the connection between these general statements and specialized symmetries, we reckon, is an interesting topic up for further investigation (we even made a similar statement about topology in our conclusion). We believe that this will be best addressed separately. • The authors illustrate their findings with the Lieb and dice lattices which both have chiral symmetry. I wonder if the authors tried to take any Hamiltonian with a flatband as a ground state and reconstruct the parent lattice? The simplest example perhaps would be the sawtooth chain. Naively that seems to be possible, since the Hamiltonian can be made positive definite, and can be decomposed into a ”square” of a rectangular matrix. However the hoppings might not be short-range again. If the answer is positive and there is a way to reconstruct the parent lattices, this would imply the hidden symmetry behind at least a subclass of flat bands. ---This is an interesting proposition. We think what the referee suggested is possible, but suspect that such a construction may not be unique. It may be unique if the starting system only has short range hoppings and then the parent system will be more directly evident (like in the case of H_5 or even what the referee stated). But on general grounds, we would consider this inverse problem to be outside the scope of current work, but as a good subsequent problem to tackle. To answer the referee’s question: we have not tried the reverse procedure as we were worried about uniqueness. • Sec. 4.1: going beyond bipartiteness – does this preserve short range hopping in general? or the long-range hoppings could be generated in this case? ---This is a good observation by the referee. In the bipartite system the hoppings generated (after projection) are at best a distance of 2 hops away in the parent system which may or may not be nn in the projected system. If we deviate from bipartiteness, the hoppings can indeed get long-ranged after projection. It isn’t a concern to us, as the parent system is still nn (which is what we wanted to preserve). In other words, we certainly don't invalidate any of the previous works but offer a simple way to arrive at them using the projection method. • It is important to compare/relate your path-exchange symmetry mechanism for the band touchings to the previously developed criterion for critical flat bands, e.g. flat bands with a touching: see ”Singular flat bands”, W Rhim, BJ Yang - Advances in Physics: X, 2021 - Taylor & Francis ---We thank the referee for pointing us to this case. Singular flatband was introduced as ones where the Bloch eigenstate of the band is discontinuous. In the case of singular flat band, when the degeneracy at the band crossing point is lifted, the band becomes dispersive and acquires a finite Chern number in general. A non-singular flat band can be isolated completely while preserving the flatness. A relevant statement for flat-band constructions is that compact localized states (CLS)-based constructions cannot be done for singular flat band systems (DOI:https://doi.org/10.1103/PhysRevB.99.045107) [although there exists some other construction addressing this with loop states]. In our work, we aren’t addressing the question of CLS or topology. In fact, the path-exchange broken system, upon projection, yields the non-singular flat bands. What is new is that, our work gives the precise prescription to generate non-singular and singular flat bands (break or don’t break path-exchange), whereas in the earlier work, only the classification was identified. This comparison/reference was needed and is now included in the conclusion as connection to previous works (lines 654-665).

---

## Round 2 · List of Changes

We corrected some minor typos and altered the variable symbols for some cases as they were used repeatedly, and added some more references to previous works. We don’t list them here. Other changes are noteworthy and are summarized below. All these changes have been discussed in the response to the referees:

  1. Couple of lines in the introduction to introduce the new sections added (see below).
  2. Lines 368-375: We added a paragraph explaining why the flat band will always be at the extremity.
  3. lines 420:447: We made minor modifications to the explanation of the symmetry and included the precise way our stated formula works.
  4. lines 478-491: We added sec 5.1 on the evolution of the gap w.r.t to the symmetry breaking parameter and the corresponding figures 11a and 11b.
  5. line 532-574: We added sec 7: about deviations from the nn considerations in our model. We discuss it for H5 and Lieb systems and quantify the effect of nnn on the gap and the bandwidth of the flat band. This section includes 4 additional figures (two for H5, one for Lieb, and one for the evolution of bandwidth) that support the claim the nnn considerations still isolate the flat band, provided we preserve the bipartite nature.
  6. lines 655-660: We added a note on possible application to lattices in hyperbolic space (with references).
  7. lines 664-675: We added some specific comparisons to previous works (Gram matrices and singular flat bands).

---

## Editorial Decision

published